# An adipokine feedback regulating diurnal food intake rhythms in mice

Anthony H Tsang[1,2†‡], Christiane E Koch[2†], Jana-Thabea Kiehn[2], Cosima X Schmidt[2], Henrik Oster[1,2*]

[1]Circadian Rhythms Group, Max Planck Institute for Biophysical Chemistry, Göttingen, Germany; [2]Institute of Neurobiology, University of Lübeck, Lübeck, Germany

**Abstract** Endogenous circadian clocks have evolved to anticipate 24 hr rhythms in environmental demands. Recent studies suggest that circadian rhythm disruption is a major risk factor for the development of metabolic disorders in humans. Conversely, alterations in energy state can disrupt circadian rhythms of behavior and physiology, creating a vicious circle of metabolic dysfunction. How peripheral energy state affects diurnal food intake, however, is still poorly understood. We here show that the adipokine adiponectin (ADIPOQ) regulates diurnal feeding rhythms through clocks in energy regulatory centers of the mediobasal hypothalamus (MBH). *Adipoq*-deficient mice show increased rest phase food intake associated with disrupted transcript rhythms of clock and appetite-regulating genes in the MBH. ADIPOQ regulates MBH clocks *via* AdipoR1-mediated upregulation of the core clock gene *Bmal1*. BMAL1, in turn, controls expression of orexigenic neuropeptide expression in the MBH. Together, these data reveal a systemic metabolic circuit to regulate central circadian clocks and energy intake.

*For correspondence:
henrik.oster@uni-luebeck.de

†These authors contributed equally to this work

Present address: ‡Wellcome Trust-MRC Institute of Metabolic Science, University of Cambridge, Cambridge, United Kingdom

Competing interests: The authors declare that no competing interests exist.

## Introduction

Worldwide, the prevalence of obesity and related disorders has reached pandemic dimensions. Besides elevated calorie-dense food intake and reduced physical activity, the timing of food intake has recently emerged as an important risk factor in this context (*Longo and Panda, 2016*). Endogenous, so called circadian clocks have evolved to anticipate daily recurring changes in the environment induced by the rotation of the Earth around its axis. Disruption of natural 24 hr rhythms, for example due to occupations requiring irregular or nighttime work hours, has been suggested as a risk factor for the development of metabolic disorders such as obesity, type-2 diabetes, and even cancer (*Garaulet et al., 2010*). On the molecular level, circadian rhythms are controlled by a genetically encoded machinery comprised of a system of interlocked transcriptional-translational feedback loops (TTFLs). During daytime, the transcriptional activator complex brain and muscle ARNT-like protein-1/circadian locomotor output cycles kaput (BMAL1/CLOCK) induces the expression of *Period (Per1-3)* and *Cryptochrome* genes (*Cry1/2*). With a delay of several hours PER/CRY complexes accumulate in the nucleus to suppress BMAL1/CLOCK function during the night. Gradual degradation of nuclear PER/CRY towards the morning releases BMAL1/CLOCK repression and reinitiates the circadian cycle (*Takahashi, 2017*). Besides *Per/Cry* expression thousands of clock-controlled genes (CCGs) are regulated in a rhythmic manner to translate the molecular clock rhythm into physiological functions (*Dibner et al., 2010*). *Via* a light responsive master pacemaker in the hypothalamic suprachiasmatic nucleus (SCN) cellular circadian clocks in central and peripheral tissues are aligned to the environmental light-dark cycle. The timing of food intake is a second potent circadian *zeitgeber* that acts primarily on peripheral tissues. Mistimed (*i.e.* rest phase) food intake uncouples peripheral clocks from the SCN (*Damiola et al., 2000*) resulting in a state of internal desynchrony which is believed to unbalance metabolic homeostasis (*Cedernaes et al., 2019a*; *Hatori et al., 2012*).

Permanent access to palatable calorie-dense diets, a hallmark of modern societies, alters diurnal rhythms of meal timing and increases food intake independent of energy demands (*Kohsaka et al., 2007*). The molecular mediators of this clock-metabolism crosstalk, however, are still poorly understood (*Cedernaes et al., 2019b*).

The MBH, especially the arcuate nucleus (ARC), houses key regulatory circuits of food intake (*Adamantidis and de Lecea, 2008*). Circulating hormones, including adipokines like ADIPOQ and leptin, convey information about the peripheral energy state to the MBH. In the ARC, these hormones modulate anorexigenic pro-opiomelanocortin (POMC)/cocaine and amphetamine-regulated transcript (CART) and orexigenic neuropeptide Y (NPY)/agouti related protein (AgRP) expressing neurons to adjust food intake. Rhythmic clock gene regulation and (an-)orexigenic neuropeptide expression (*Fick et al., 2010*; *Guilding et al., 2009*) are dampened in mice with disrupted feeding rhythms indicating a crucial impact of circadian MBH rhythms in food intake regulation (*Kohsaka et al., 2007*). Furthermore, some of the central acting appetite-regulating hormones show pronounced circadian rhythms. Therefore, we postulated that peripheral metabolic hormones may reset MBH clocks to adjust circadian appetite regulation in response to changes in the body's energy state.

Under normal metabolic conditions ADIPOQ is one of the most abundant hormones in the blood with potent anti-inflammatory, insulin sensitizing, and appetite-regulatory properties (*Koch et al., 2014*; *Yamauchi and Kadowaki, 2013*). ADIPOQ receptor and receptor target gene expression in the MBH shows circadian rhythmicity (*Cedernaes et al., 2019a*; *Kohsaka et al., 2007*; *Zhang et al., 2014*). Moreover, in obese patients, ADIPOQ blood levels decline and diurnal rhythms of ADIPOQ release are dampened (*Calvani et al., 2004*; *Yildiz et al., 2004*). These data led us to hypothesize that ADIPOQ acts as a mediator between energy state and central appetite regulation *via* resetting of MBH circadian clocks.

## Results

### ADIPOQ signaling reflects systemic metabolic state

We first investigated the regulation of ADIPOQ signaling under different metabolic conditions in wild-type (WT) male mice. Under *ad-libitum* access to normal chow (NC) diet, *Adipoq* mRNA levels in epididymal white adipose tissue (eWAT) showed robust diurnal rhythms peaking around the day-night transition (*Figure 1A*; *Barnea et al., 2015*). With a delay of a few hours this rhythm was followed by ADIPOQ protein levels in plasma (*Figure 1B*). Interestingly, transcript levels of the two ADIPOQ receptors, *Adipor1/2*, in the MBH also showed robust, but antiphasic, rhythms under these conditions with *Adipor1* mRNA peaking at the beginning and *Adipor2* at the end of the light phase (*Figure 1C*). In line with a rhythmic ADIPOQ signal peaking at the day-night transition, the expression of two ADIPOQ receptor target genes and appetite regulators, *Npy* and *Agrp*, was also rhythmic, peaking at the end of the day and coinciding with the beginning of the normal feeding phase (*Figure 1D*). Expression levels of two anorexigenic markers, *Pomc* and *Cartpt*, were moderately rhythmic with higher levels during the dark phase (*Figure 1E*).

In order to act as a feedback signal of peripheral energy state to the brain, adiponectin signaling would be expected to change under altered metabolic conditions. To test this, we subjected mice to 4 hr time restricted feeding regimens with food access either at *zeitgeber* time (ZT) 7–11 (light phase restricted feeding - L-RF) or ZT19-23 (dark phase RF - D-RF) for two weeks. Under L-RF conditions, *Adipoq* mRNA rhythms in eWAT were dampened, while under D-RF conditions, the mRNA peak was shifted into the night phase. Of note, under *ad-libitum* and RF conditions, *Adipoq* mRNA levels always peaked at the onset of the feeding phase, that is at ZT12 under *ad-libitum*, at ZT6 under L-RF, and at ZT18 under D-RF conditions (*Figure 1F*). Similar effects were observed in plasma with ADIPOQ peaking at noon under L-RF and around midnight under D-RF conditions (*Figure 1G*). After some days of adaptation total energy intake was not significantly altered in RF compared to *ad-libitum* conditions (4.5 ± 0.2 g (ad libitum) *vs.* 4.3 ± 0.3 g (L-RF) *vs.* 4.4 ± 0.2 g (D-RF)). Since feeding rhythms and, thus, metabolic state oscillations were most profoundly changed under L-RF conditions when compared to the *ad-libitum* situation, we focused on this condition to study ADIPOQ signaling in the MBH. The diurnal expression rhythm of *Adipor1* mRNA in the MBH was dampened under L-RF conditions, while the peak of *Adipor2* mRNA was shifted into the day (*Figure 1H*). In line

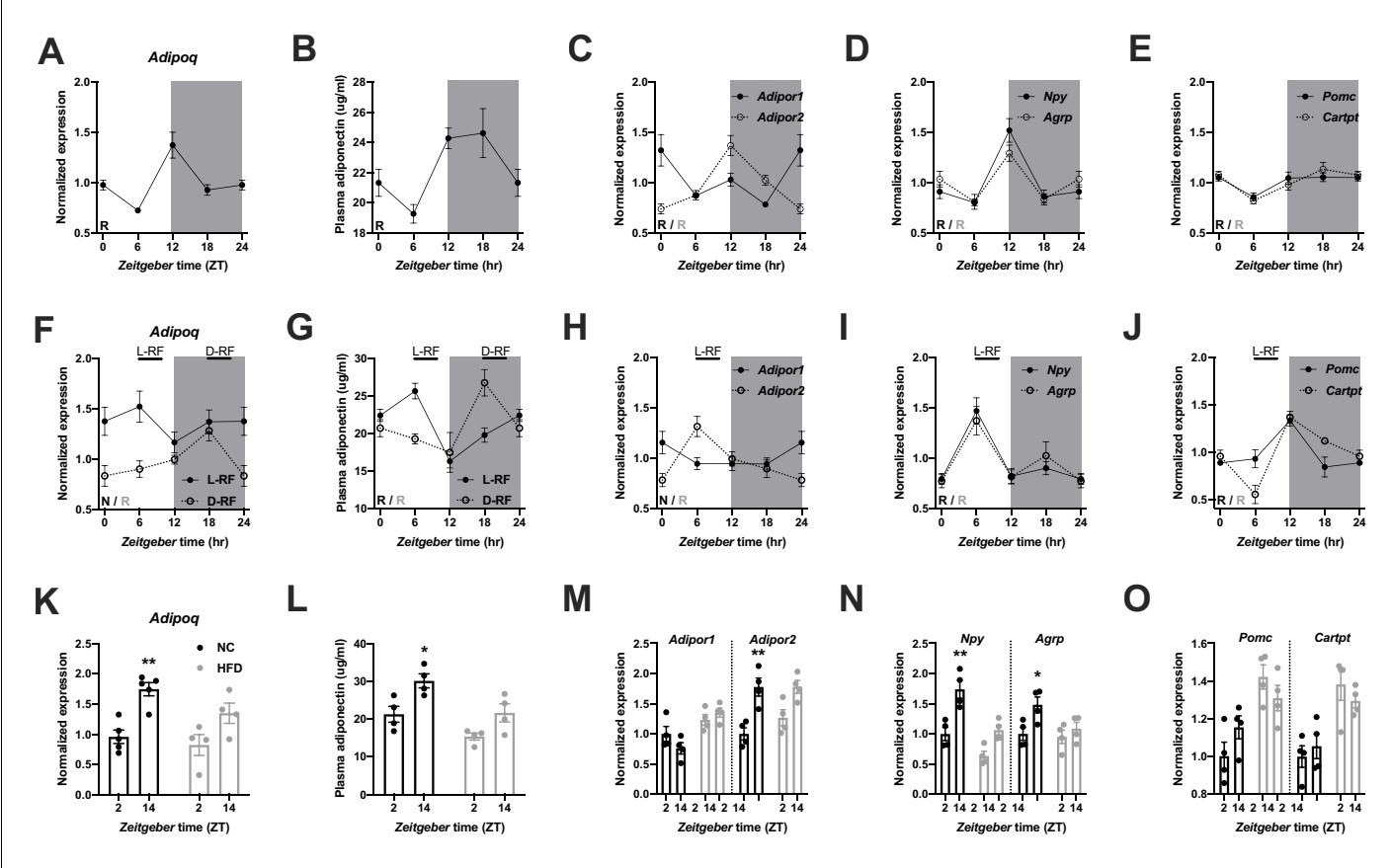

**Figure 1.** Adiponectin signaling integrates energy metabolic state in male mice. (**A**) Daily profile of *Adipoq* mRNA expression in epididymal white adipose tissue (eWAT; n = 5 per time point). (**B**) Daily profile of ADIPOQ peptide in plasma (n = 3 per time point). (**C**) Daily profiles of *Adipor1* (filled circles) and −2 (open circles) mRNA expression in the mediobasal hypothalamus (MBH; n = 5 per time point). (**D, E**) Daily profiles of *Npy, Agrp* (**D**), *Pomc* and *Cartpt* (**E**) mRNA expression in the MBH (n = 5 per time point). (**F, G**) Daily profiles of eWAT *Adipoq* mRNA expression (**F**) and plasma ADIPOQ levels (**G**) under light phase restricted feeding (L-RF; filled circles) or dark phase restricted feeding conditions (D-RF; open circles; n = 3 per time point). (**H–J**) Daily profiles of MBH *Adipor1/2* (**H**), *Npy/Agrp* (**I**) and *Pomc/Cartpt* (**J**) mRNA expression under L-RF conditions (n = 3 per time point). (**K, L**) Time of day differences in eWAT *Adipoq* mRNA expression (**K**) and plasma ADIPOQ levels (**L**) under normal chow (NC; black) and after 1 week of high-fat diet (HFD; grey) conditions (n = 4). (**M–O**) Time of day differences in MBH mRNA expression of *Adipor1/2* (**M**), *Npy/Agrp* (**N**) and *Pomc/Cartpt* (**O**) under NC and HFD conditions (n = 4). All data are means ± SEM. R – rhythmic expression (p<0.05; CircWave); N – non-rhythmic expression (p>0.05); */** - p<0.05/0.01; 2-way ANOVA with Sidak's multiple comparisons. p values: (**K**) 0.005 (NC; ZT2 *vs.* ZT14), ANOVA dF = $F(1, 14)$=0.90 (interaction)/22.63 (time)/3.73 (diet); (**L**) 0.029 (NC; ZT2 *vs.* ZT14), ANOVA dF = $F(1, 12)$=0.45 (interaction)/15.66 (time)/14.18 (diet); (**M**) 0.001 (*Adipor2*; NC; ZT2 *vs.* ZT14), ANOVA dF = $F(3, 24)$=7.98 (interaction)/13.74 (time)/12.35 (diet); (**N**) 0.002 (*Npy*; NC; ZT2 vs. ZT14) and 0.044 (*Agrp*; NC; ZT2 *vs.* ZT14), ANOVA dF = $F(3, 24)$=2.51 (interaction)/32.44 (time)/8.70 (diet).

The online version of this article includes the following source data for figure 1:

**Source data 1.** Raw data of experiments shown in *Figure 1*.

with this, *Npy* and *Agrp* mRNA rhythms were phase-advanced in L-RF with peak expression at ZT6 (*Figure 1I*) while *Pomc/Cartpt* expression peaked shortly after the feeding interval at ZT12 (*Figure 1J*).

It has previously been shown that circadian rhythms of hypothalamic gene expression and of food intake are strongly dampened under high-fat diet (HFD) conditions (*Kohsaka et al., 2007*). This was reflected at the level of ADIPOQ signaling. In eWAT, changes in *Adipoq* mRNA expression between morning (ZT2) and evening (ZT14) were reduced after one week under HFD conditions (*Figure 1K*). Similar effects were seen for plasma ADIPOQ (*Figure 1L*). In the MBH, expression of *Adipor1* did not differ between timepoints under chow conditions and was slightly upregulated on HFD while *Adipor2* regulation was largely comparable between both conditions with higher expression at ZT14. However, this upregulation was no longer significant under HFD conditions (*Figure 1M*). *Npy/*

*Agrp* mRNA expression in the MBH was upregulated at ZT14 under chow conditions and this effect was lost under HFD (*Figure 1N*). *Pomc/Cartpt* expression was upregulated under HFD conditions without marked differences between timepoints (*Kohsaka et al., 2007*; *Figure 1O*). Overall, these data suggest that ADIPOQ signaling is responsive to daily alterations in metabolic state.

## Dampened hypothalamic and feeding rhythms in *Adipoq*-deficient mice

To investigate the physiological relevance of ADIPOQ signaling in vivo, we analyzed rhythmic behavior and gene expression in ADIPOQ-deficient mice (MT) (*Ma et al., 2002*). Under rhythmic light-dark (LD) conditions, activity patterns were largely comparable between male WT and congenic, age-matched MT mice with little activity during the light phase and maximal activity at ZT12-14, but MT mice showed higher activity during the first hours of the dark phase (*Figure 2A*, *Figure 2—figure supplement 1A–C*). At the same time, however, the diurnal rhythm of food intake was dampened in mutant mice with a ~ 50% increase in relative light phase food intake (19.8 ± 1.3% *vs.* 29.4 ± 2.3%; *Figure 2B,C*) while total energy intake was unaltered (*Figure 2—figure supplement 1D*). In the absence of external time information in constant darkness (DD) these findings were preserved (and even more pronounced than in LD), indicating a regulation by the endogenous circadian clock system (*Figure 2D–F*, *Figure 2—figure supplement 1A,E–G*). Of note, DD activity free-running period lengths were comparable between both genotypes indicating that the SCN pacemaker was unaffected by the loss of ADIPOQ (*Figure 2—figure supplement 1A,F*).

The changes in feeding rhythmicity were reminiscent of what had previously been observed in mice under HFD conditions (*Kohsaka et al., 2007*) and clock gene mutant animals (*Cedernaes et al., 2019a*; *Kohsaka et al., 2007*). In both cases behavioral rhythm dampening was accompanied by a similar dampening in hypothalamic gene expression. Such effect was also observed in MT mice with altered and blunted clock and metabolic gene mRNA rhythms in the MBH (*Figure 2G–L*). Interestingly, these clock effects were tissue-specific since no changes in clock gene mRNA profiles were seen in SCN, liver, and skeletal muscle (*Figure 2—figure supplement 2*).

We repeated the experiments in female WT and *Adipoq* deficient mice to test for gender-specific effects of ADIPOQ on activity and appetite regulation. LD activity patterns were comparable between female WT and MT mice with little activity during the light phase and maximal activity at ZT12-14 (*Figure 3A*, *Figure 3—figure supplement 1A–C*). Similar to males, the diurnal rhythm of food intake was markedly dampened in female mutant mice with a ~ 50% increase in relative light phase food intake (17.3 ± 4.6% *vs.* 37.7 ± 6.8%; *Figure 3B,C*) while total energy intake was unaltered (*Figure 3—figure supplement 1D*). In DD, these findings were preserved (*Figure 3D–F*, *Figure 3—figure supplement 1A,E–G*). Also, DD activity free-running period lengths were comparable between both genotypes (*Figure 3—figure supplement 1A,F*).

Dampening of gene expression profiles was also observed in the MBH of female MT mice (*Figure 3G–L*), but not in SCN, liver, and skeletal muscle (*Figure 3—figure supplement 2*). In summary, the MBH gene expression/feeding rhythm phenotype observed in male *Adipoq* mutant mice was preserved in females.

## Rescue of diurnal feeding rhythms by timed ADIPOQ treatment

To further confirm the relevance of ADIPOQ for the timing of food intake, we rescued ADIPOQ deficiency centrally in *Adipoq* mutant mice by timed *i.c.v.* infusion of an ADIPOQ receptor agonist (AdipoRon; *Figure 4A*; *Okada-Iwabu et al., 2013*). While constant activation of central ADIPOQ signaling did not improve the feeding rhythm of MT mice, timed infusion only during the dark phase normalized the diurnal feeding rhythm of the mutants to that of WT animals (*Figure 4B,C*). Of note, regardless of the timing of infusion, total food consumption was not significantly altered (*Figure 4—figure supplement 1*). In line with the observed normalization of food intake rhythms, timed AdipoRon also restored temporal variations in expression of the clock gene *Bmal1* and the orexigenic genes *Npy* and *Agrp* in the MBH of MT mice (*Figure 4D–F*). In sum, these data demonstrate the relevance of daily variations of central ADIPOQ signaling for the diurnal regulation of feeding behavior.

## ADIPOQ signaling in the MBH

To understand how the timed activation of the ADIPOQ signaling cascade normalizes rhythmic gene expression and food intake in the MBH, we investigated the acute impact of ADIPOQ on

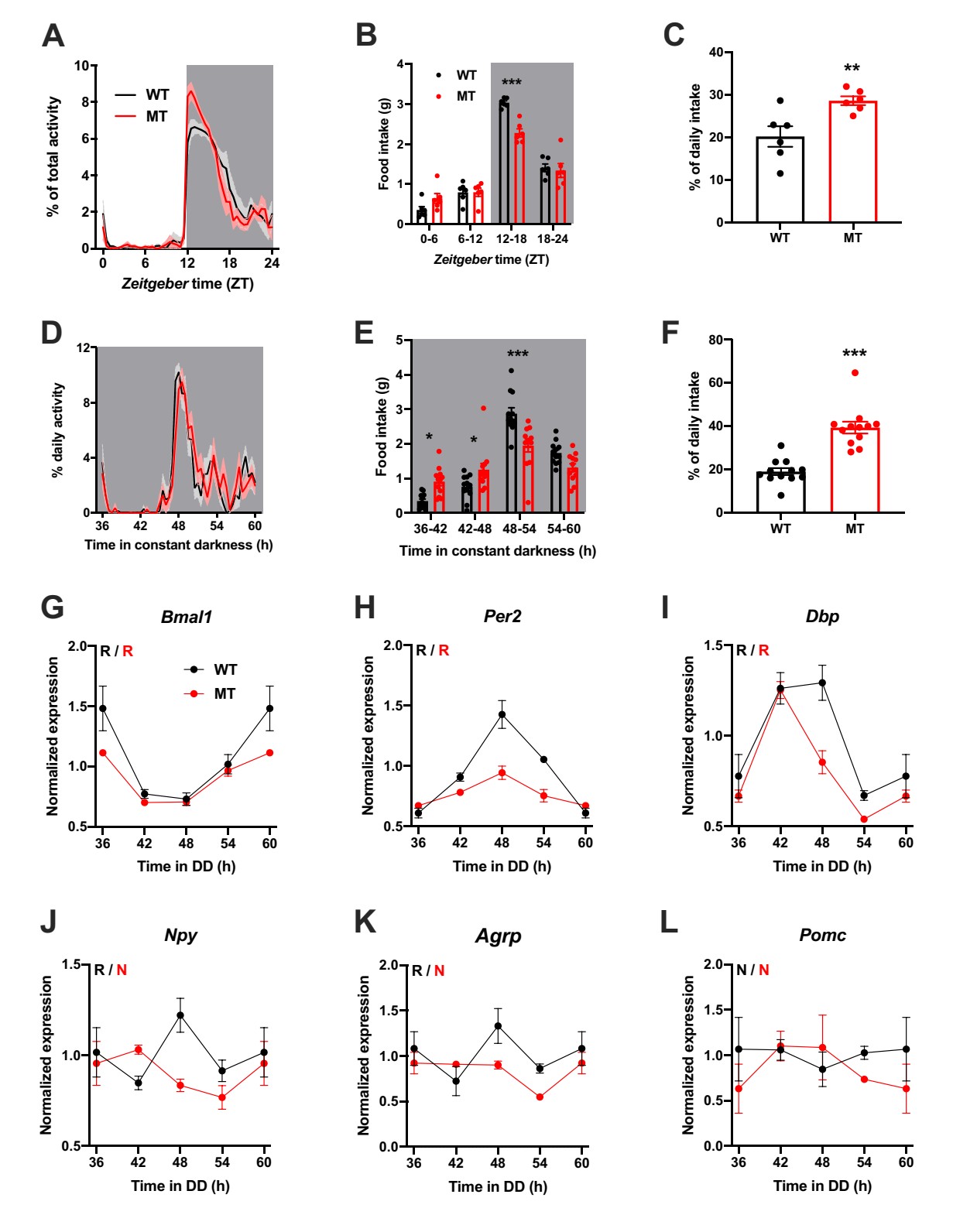

**Figure 2.** Dampened circadian rhythms of food intake and hypothalamic clock gene expression in *Adipoq*-deficient male mice. (**A**) Daily profiles of running-wheel activity of wild-type (WT; black) and *Adipoq*-deficient mice (MT; red) under 12 hr light: 12 hr dark (LD) conditions (n = 6 per group; grey shading indicates dark phase). (**B**) Daily food intake profiles of WT and MT mice in LD in 6 hr bins (n = 6 per group). (**C**) Relative light phase food intake of WT and MT mice in LD (n = 6). (**D–F**) circadian profiles of running-wheel activity (D; n = 12 per group), food intake (E; n = 12 per group) and relative

*Figure 2 continued on next page*

*Figure 2 continued*

rest phase food intake (F) of WT and MT mice in constant darkness (DD; n = 12 per group). (G–L) Circadian profiles of MBH clock gene (G–I) and metabolic gene (J–L) mRNA expression in WT and MT mice in DD (n = 3 per time point). All data are means ± SEM. R – rhythmic expression (p<0.05; CircWave); N – non-rhythmic expression (p>0.05); */**/*** - p<0.05/0.01/0.001; 2-way ANOVA with Sidak's multiple comparisons for B, E; unpaired Student's t-test for C, F. p-values: (B) < 0.0001 (MT *vs.* WT ZT12-18), ANOVA dF = $F(3, 40)$=8.32 (interaction)/155.4 (time)/3.03 (genotype); (C) 0.0097 (MT *vs.* WT); (E) 0.015 (MT *vs.* WT 36–42), 0.040 (42-48),<0.0001 (48-54), ANOVA dF = $F(3, 88)$=14.79 (interaction)/66.95 (time)/0.61 (genotype); (F) < 0.0001 (MT *vs.* WT).

The online version of this article includes the following source data and figure supplement(s) for figure 2:

**Source data 1.** Raw data of experiments shown in *Figure 2*.
**Figure supplement 1.** Normal rest-activity rhythms in *Adipoq*-deficient male mice.
**Figure supplement 2.** Preserved circadian clock gene expression rhythms in the SCN and peripheral tissues of *Adipoq*-deficient male mice.

hypothalamic neuronal clocks in vitro. mHypo-N44 (N44) cells of murine hypothalamic origin were stably transduced with a lentiviral construct expressing a circadian *Bmal1-luc* reporter (*Brown et al., 2008*). After synchronization with dexamethasone, N44 *Bmal1-luc* cells showed robust rhythms of luciferase activity for more than 5 d in culture (*Figure 5A*). Synchronized cells were treated with ADIPOQ at different phases of the luminescence rhythm, yielding phase-dependent shifts between −4 and +4 hr (*Figure 5B,C*). When cells were treated with different doses of ADIPOQ at the cardinal points of the phase-response curve (PRC), dose effects were observed for both directions (*Figure 5— figure supplement 1*). ADIPOQ treatment also had treatment time-dependent phase-shifting effects in primary MBH neurons (*Figure 5D*) and organotypic MBH slices of *PER2::LUC* circadian reporter mice (*Figure 5E,F*; *Yoo et al., 2004*). Of note, no phase shifts of N44 *Bmal1-luc* cells were observed after treatment with several other adipokines and metabolic peptides that had previously been implicated in circadian clock regulation such as leptin, visfatin, insulin, ghrelin, glucagon, and resistin, demonstrating the specificity of the assay (*Figure 5—figure supplement 2*). The shape of the ADI-POQ PRC of N44 *Bmal1-luc* cells was markedly different from those of two well-known clock regulators acting through activation of *Per* gene expression, the glucocorticoid analogue dexamethasone (DEX) and the adenylate cyclase activator forskolin (Fors; *Figure 5—figure supplement 3*). These data suggested that ADIPOQ affects the clock through a different pathway.

When treating N44 *Bmal1-luc* cells with ADIPOQ, we consistently observed an upregulation of luciferase activity in the hours directly after the treatment. To confirm this, we treated N44 *Bmal1-luc* cells with ADIPOQ and monitored *Bmal1-luc* activity during the following 24 hr. Indeed, luminescence was strongly upregulated in ADIPOQ- compared to PBS-treated unsynchronized (*Figure 6A*) and synchronized cells (*Figure 6—figure supplement 1A,B*). In the latter, the activation of *Bmal1* expression was confirmed at the levels of *Bmal1* mRNA and BMAL1 protein (*Figure 6—figure supplement 1C*). *Bmal1* mRNA induction was similarly observed in the MBH of WT mice 3 hr after *i.v.* injection of ADIPOQ at ZT6 (*Figure 6B*) indicating that peripheral ADIPOQ can cross the blood-brain barrier to affect circadian clock function in the hypothalamus. Expression of the clock genes *Per1/2/3* and *Rev-erb alpha* (*Nr1d1*) was not significantly altered neither in N44 cells nor in the MBH (*Figure 6C–F*). An upregulation of gene expression after ADIPOQ treatment similar to *Bmal1*, however, was observed in cells and in the hypothalamus of treated mice for *Npy* and *Agrp* (*Figure 6G, H*). In *Bmal1*-deficient N44 cells and in the MBH of *Bmal1* knockout mice, this ADIPOQ-mediated induction of *Npy/Agrp* was blunted (*Figure 6I,J*). Similarly, *Per2* was not regulated by ADIPOQ treatment (*Figure 6K*). Together, these data suggest that *Bmal1* may mediate the effect of ADIPOQ on metabolic peptide gene regulation. In HEK293T cells, we tested the direct regulation of *Npy* transcription by the CLOCK/BMAL1 heterodimer. Increasing doses of *Clock/Bmal1* expression vector led to higher levels of luciferase expressed from an *Npy-luc* reporter construct. This induction was reduced by addition of the *E-box* repressor *Cry1* (*Figure 6L*). Together, these data suggest that *Npy* is a *bona fide* clock-controlled gene which is affected by ADIPOQ through *Bmal1*.

Using an shRNA approach, we knocked down (KD) the two ADIPOQ receptors, *Adipor1* and *-r2*, individually to less than one fifth of their original levels in N44 cells (*Figure 6M*). Interestingly, KD of *Adipor1* led to concomitant downregulation of *Adipor2* mRNA of about 50%, while KD of *Adipor2* had no effect on the expression of *Adipor1* (*Figure 6M*). KD of *Adipor1* and of both receptors simultaneously led to a reduction of *Bmal1* and *Npy* expression and of ADIPOQ-induced phase shifts in N44 cells (*Figure 6N–P*). Surprisingly, KD of *Adipor2* alone had the opposite effect on *Bmal1* and

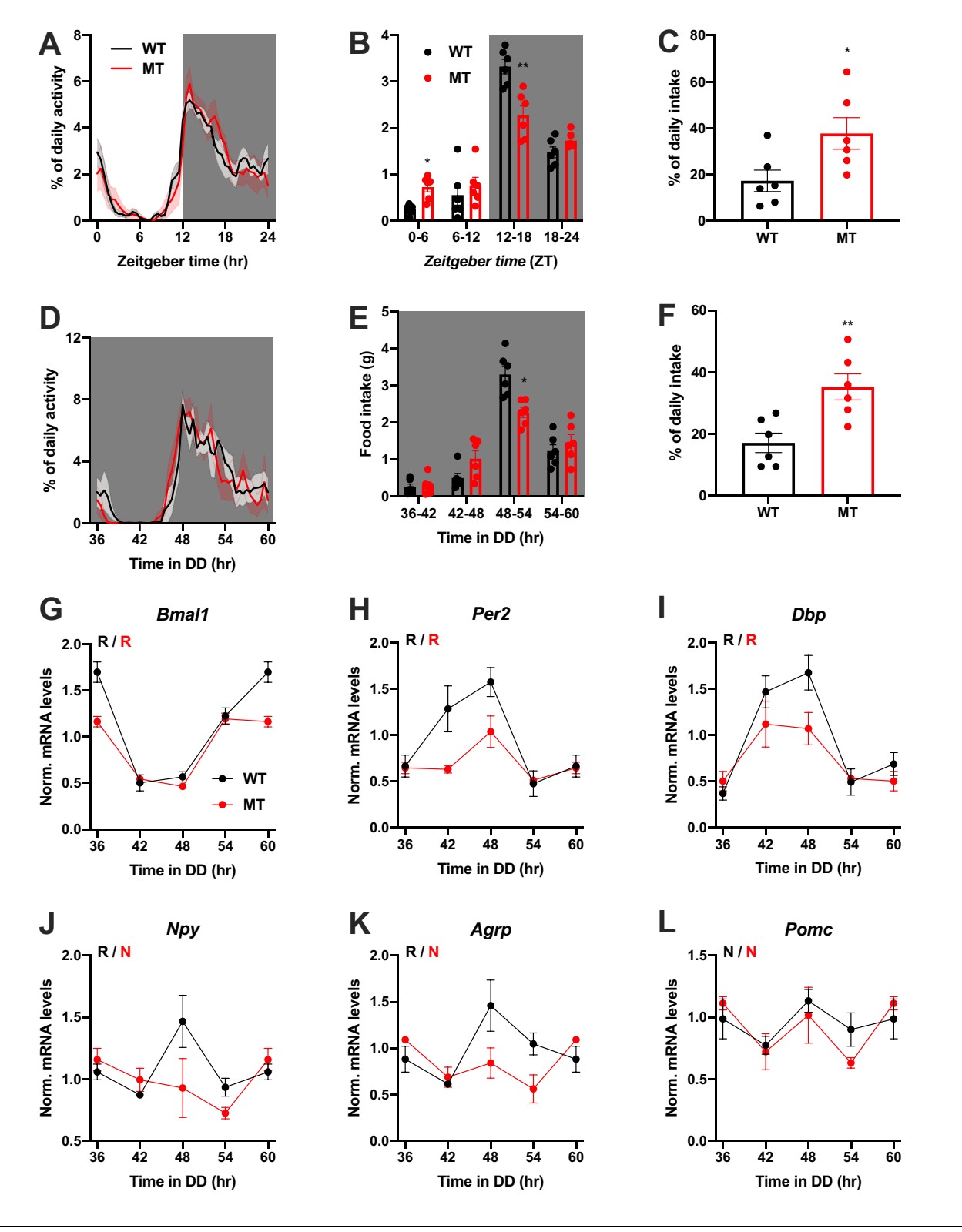

**Figure 3.** Dampened circadian rhythms of food intake and hypothalamic clock gene expression in *Adipoq*-deficient female mice. (**A**) Daily profiles of running-wheel activity of wild-type (WT; black) and *Adipoq*-deficient mice (MT; red) under 12 hr light: 12 hr dark (LD) conditions (n = 6 per group; grey shading indicates dark phase). (**B**) Daily food intake profiles of WT and MT mice in LD in 6 hr bins (n = 6 per group). (**C**) Relative light phase food intake of WT and MT mice in LD (n = 6 per group). (**D–F**) Circadian profiles of running-wheel activity (D; n = 6 per group), food intake (E; n = 6 per group) and

*Figure 3 continued on next page*

*Figure 3 continued*

relative rest phase food intake (F) of WT and MT mice in constant darkness (DD; n = 6 per group). (G–L) Circadian profiles of MBH clock gene (G–I) and metabolic gene (J–L) mRNA expression in WT and MT mice in DD (n = 3 per time point). All data are means ± SEM. R – rhythmic expression (p<0.05; CircWave); N – non-rhythmic expression (p>0.05); */**/*** - p<0.05/0.01/0.001; 2-way ANOVA with Sidak's multiple comparisons for B, E; unpaired Student's t-test for C, F. p-values: (B) 0.0145 (MT *vs.* WT ZT0-6) and 0.0097 (MT *vs.* WT ZT12-18), ANOVA dF = $F_{(3, 30)}$=10.21 (interaction)/95.89 (time)/ 0.10 (genotype); (C) 0.0325 (MT *vs.* WT); (E) 0.0206 (MT *vs.* WT 48–54), ANOVA dF = $F_{(3, 30)}$=7.63 (interaction)/79.32 (time)/0.30 (genotype); (F) 0.0062 (MT *vs.* WT).

The online version of this article includes the following source data and figure supplement(s) for figure 3:

**Source data 1.** Raw data of experiments shown in *Figure 3*.

**Figure supplement 1.** Normal rest-activity rhythms in *Adipoq*-deficient female mice.

**Figure supplement 2.** Preserved circadian clock gene expression rhythms in the SCN and peripheral tissues of *Adipoq*-deficient female mice.

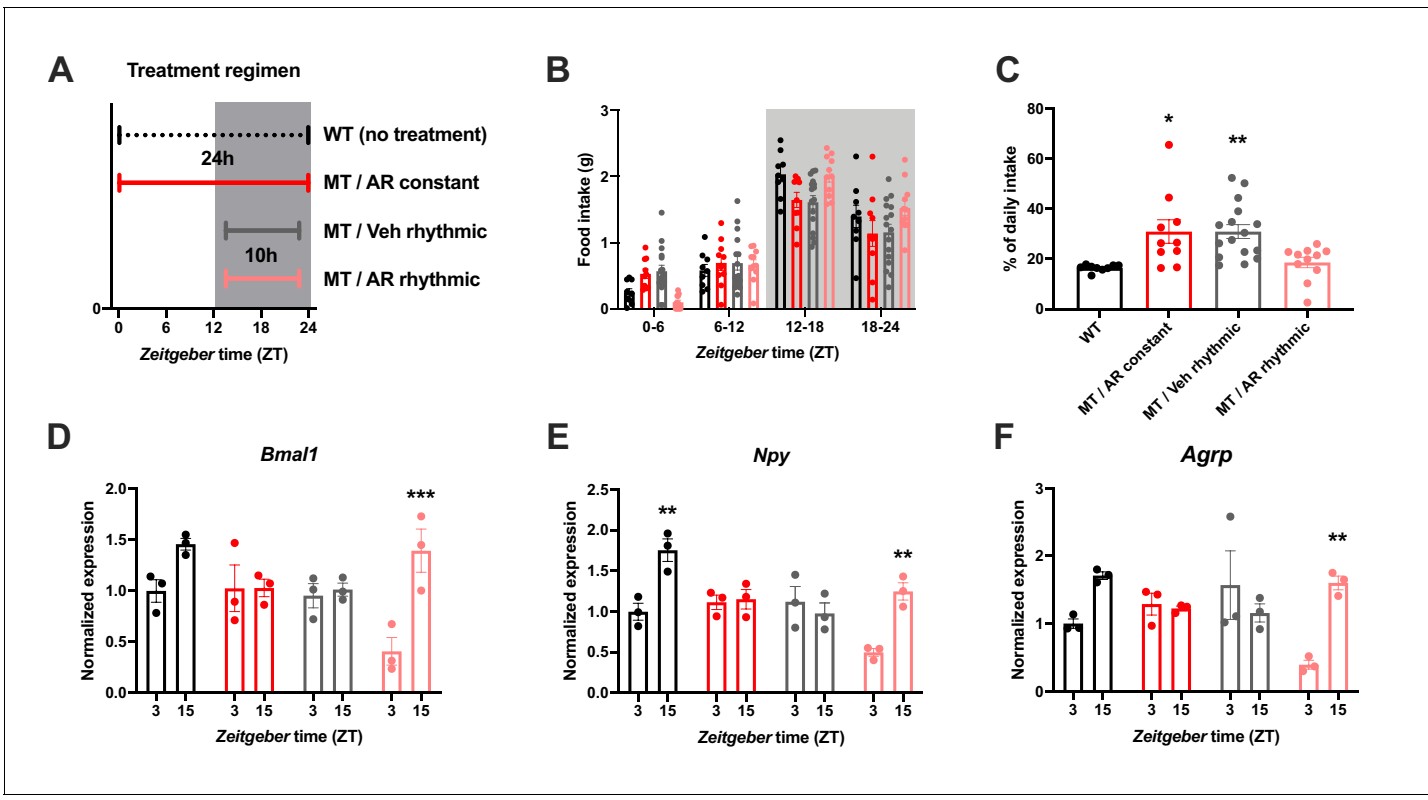

**Figure 4.** Restoration of food intake rhythms by rhythmic activation of central adiponectin signaling in male mice. (A) Treatment regimen sketch. Grey shading indicates dark phase. (B) Daily food intake profiles of untreated wild-type mice (WT; black; n = 9), *Adipoq*-deficient mice with constant AdipoRon *i.c.v.* infusion (MT/AR constant; red; n = 10), MT mice with rhythmic (10 hr per day) vehicle infusion (MT/Veh rhythmic; grey; n = 16), and MT mice with rhythmic AdipoRon infusion (MT/AR rhythmic; pink; n = 11). (C) Relative light phase food intake of WT mice and MT mice with constant or rhythmic AdipoRon or vehicle *i.c.v* infusion. (D–F) Time of day differences in MBH mRNA expression of *Bmal1* (D), *Npy* (E) and *Agrp* (F) in WT and MT mice with constant (red) or rhythmic *i.c.v.* infusion of AdipoRon (pink) or vehicle (grey) (n = 3 per time point). All data are means ± SEM. */**/*** - p<0.05/0.01/0.001; 1-way ANOVA (C) or 2-way ANOVA with Sidak's multiple comparisons (D–F). p-values: (C) 0.017 (MT/AR constant *vs.* WT), 0.007 (MT/ Veh rhythmic *vs.* WT), ANOVA dF = $F_{(3, 42)}$=6.56; (D) 0.0005 (AR rhythmic, ZT3 *vs.* ZT15), ANOVA dF = $F_{(3, 16)}$=30.24 (interaction)/27.8 (time)/11.37 (treatment); (E) 0.0019 (WT, ZT3 *vs.* ZT15) and 0.0017 (AR rhythmic, ZT3 *vs.* ZT15), ANOVA dF = $F_{(3, 16)}$=30.98 (interaction)/22.76 (time)/24.39 (treatment); (F) 0.0024 (AR rhythmic, ZT3 *vs.* ZT15), ANOVA dF = $F_{(3, 16)}$=42.73 (interaction)/13.77 (time)/9.25 (treatment).

The online version of this article includes the following source data and figure supplement(s) for figure 4:

**Source data 1.** Raw data of experiments shown in *Figure 4*.

**Figure supplement 1.** Central AdipoRon treatment has no effect on total food intake in *Adipoq*-deficient male mice.

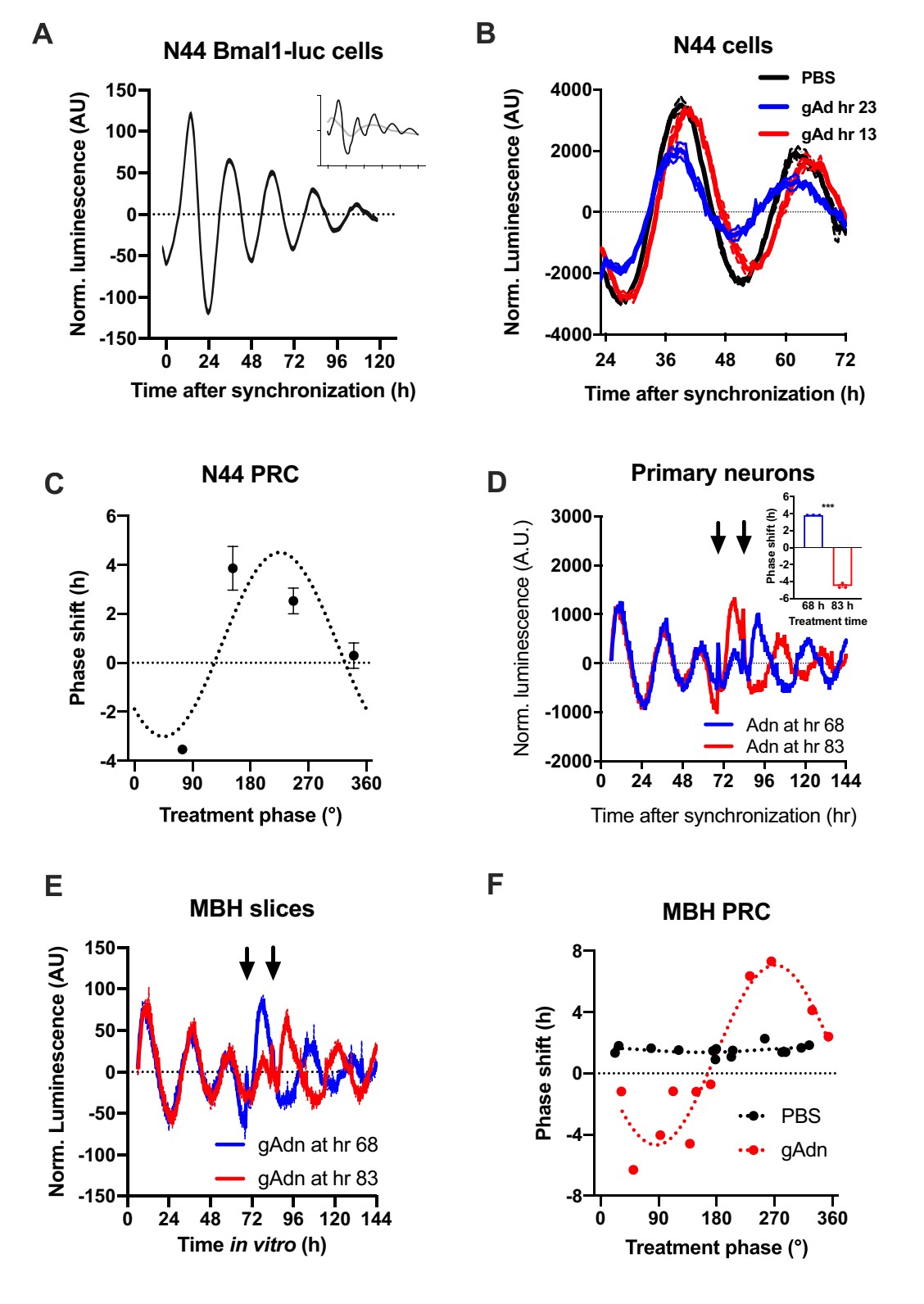

**Figure 5.** Resetting of hypothalamic clocks by ADIPOQ. (**A**) Representative normalized *Bmal1-luc* luminescence rhythms of dexamethasone (DEX) synchronized hypothalamus-derived mHypo-N44 (**N44**) cells. Inset depicts raw luminescence data of the same set. (**B**) Normalized *Bmal1-luc* luminescence rhythms of N44 cells after treatment with ADIPOQ (Adn; red and blue) peptide or PBS (black) at the depicted time points after synchronization. Shown are averages ± SEM of 3 traces each. (**C**) Phase response curve for ADIPOQ-mediated resetting of N44/*Bmal1-luc* cells (n = 3
*Figure 5 continued on next page*

Figure 5 continued

per time point; treatment time given in degrees with 90 °=maximal luminescence and 270 °=minimal luminescence – see (**A**) for reference). (**D**) Normalized *PER2::LUC* luminescence rhythms of primary hypothalamic neurons after treatment with ADIPOQ (Adn; red and blue) peptide at the depicted time points after synchronization. Shown are averages ± SEM of 3 traces each. Inset shows quantification of phase shifts (p<0.0001; unpaired Student's t-test). (**E**) Normalized *PER2::LUC* luminescence rhythms of organotypic MBH slices after treatment with ADIPOQ (Adn; red and blue) peptide at the depicted time points. Shown are averages ± SEM of 3 traces each. (**F**) Phase response curve for ADIPOQ-mediated resetting of *PER2::LUC* MBH slices (treatment time given in degrees with 90 °=maximal luminescence and 270 °=minimal luminescence – see (**E**) for reference.

The online version of this article includes the following source data and figure supplement(s) for figure 5:

**Source data 1.** Raw data of experiments shown in *Figure 5*.
**Figure supplement 1.** ADIPOQ resets hypothalamic clocks in a dose-dependent manner.
**Figure supplement 2.** Hormonal resetting of clocks in N44/*Bmal1-luc* cells.
**Figure supplement 3.** Resetting of clocks in N44/*Bmal1-luc* cells by dexamethasone and forskolin.

*Npy* expression, but did not affect ADIPOQ-induced rhythm resetting in N44 *Bmal1-luc* cells (*Figure 6N–P*).

The in-vitro data suggested that ADIPOR1 is the main mediator of ADIPOQ's clock- and appetite-resetting effects. To test this, we knocked down *Adipor1* expression in the MBH by stereotaxic injection of an *Adipor1* shRNA expressing AAV. *Adipor1* KD led to a 60% reduction in *Adipor1* mRNA and a 70% reduction of ADIPOR1 protein in the arcuate nucleus (ARC) of WT mice (*Figure 6Q*). ADIPOR1 KD did not affect cumulative 24 hr food intake, but led to a dampened feeding rhythm with elevated light phase intake under LD and DD conditions (*Figure 6*, R-T) similar to what had been observed in *Adipoq^{-/-}* mice (*Figures 2* and *3*) and mice fed an HFD (*Kohsaka et al., 2007*).

qPCR analysis in the MBH revealed that expression of a known downstream target of ADIPOR1 (*Iwabu et al., 2010*) and regulator of *Bmal1* transcription (*Liu et al., 2007*), peroxisome proliferator-activated receptor gamma coactivator 1-alpha (*Pgc1a* or *Ppargc1a*), was rhythmic in WT and non-rhythmic in MT mice (*Figure 7A*). Treatment with ADIPOQ induced *Pgc1a* expression in N44 cells and – though not significantly – in the MBH of WT mice (*Figure 7B*) while knock-down of *Adipor2* had the opposite effect (*Figure 7C*). To test if PGC1a is involved in mediating ADIPOQ's effects on hypothalamic clocks we treated N44 cells with ADIPOQ with or without *prior* knock-down of *Pgc1a*. Treatment with *Pgc1a* shRNA resulted in a slight period lengthening together with a 1.5 hr phase advance and ca. 30% increased dampening rates in synchronized N44 cells (*Figure 7—figure supplement 1*). In unsynchronized cells, shRNA transfection resulted in a *Pgc1a* knock-down efficiency of 60% at the mRNA level, concomitant with a reduction in *Bmal1* transcript levels of about 40% (*Figure 7D*). Knock-down of *Pgc1a* inhibited ADIPOQ induced upregulation of *Bmal1* (*Figure 7E*). Similarly, inhibition of the PGC1a partner and *Bmal1* regulator retinoid orphan receptor alpha (RORa) (*Liu et al., 2007*) by treatment with VPR66 blunted ADIPOQ-induced mRNA expression of *Bmal1* (*Figure 7F*). Chromatin immunoprecipitation (ChIP) of PGC1a followed by qPCR from ADIPOQ-treated N44 cells showed ADIPOQ-induced binding of PGC1a to *Bmal1 ROR* elements (*Figure 7G*). Finally, knock-down of *Pgc1a* inhibited ADIPOQ-induced phase shifts of *Bmal1*-luciferase luminescence rhythms in synchronized N44 cells (*Figure 7H,I*). In summary, our data suggest that ADIPOQ may act *via* an ADIPOR1-PGC1a-BMAL1 pathway to reset clocks in hypothalamic neurons.

## Timed activation of central ADIPOQ signaling improves diet-induced obesity

So far, our data point at a role for ADIPOQ signaling in the central regulation of diurnal appetite rhythms. Considering the weight-promoting effects of rest phase food intake (*Arble et al., 2009*) and the therapeutic potential of time-restricted eating for diet-induced obesity (*Hatori et al., 2012*), we speculated that timed activation of hypothalamic ADIPOQ signaling may improve body weight regulation under obesogenic conditions. We therefore treated obese WT mice either constantly or during the dark phase (10 hr during every night; ZT13-ZT23) with AdipoRon and monitored food intake and body weight regulation (*Figure 8A*). Timed central AdipoRon infusion was started after 6–7 weeks of HFD using programmable implanted mini pumps. Already after one week of treatment, diurnal food intake rhythms of HFD mice were improved by rhythmic but not constant AdipoRon

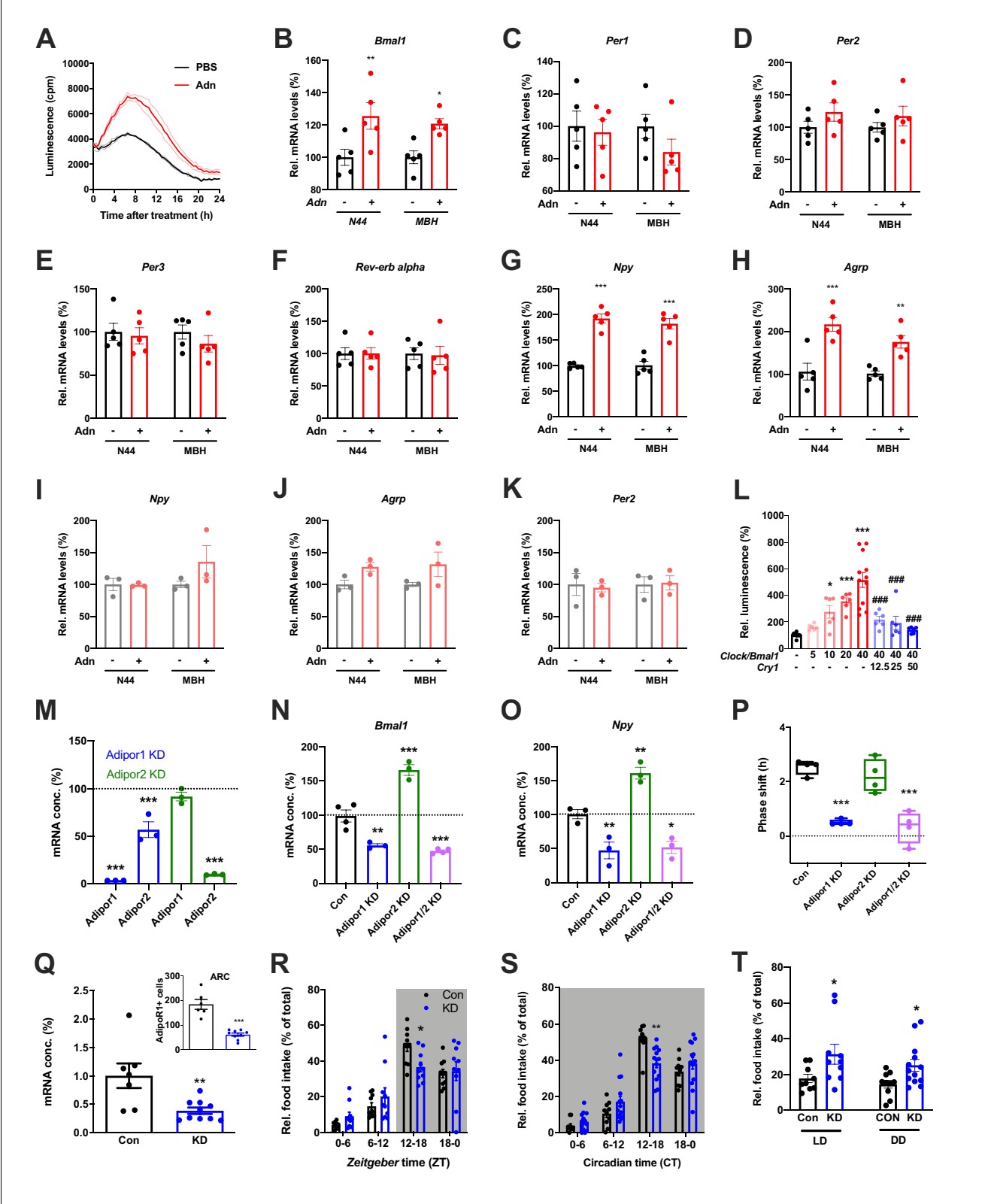

**Figure 6.** Adiponectin signaling modulates hypothalamic appetite regulation in male mice. (**A**) Transient upregulation of *Bmal1-luc* activity in non-synchronized N44 cells after adiponectin (Adn) treatment. Shown are averages ± SEM of 3 traces each. (**B–H**) Induction of mRNA levels in N44 cells after Adn treatment and in the mediobasal hypothalamus (MBH) after *i.v.* Adn injection in wild-type mice; (**B**) *Bmal1*, (**C**) *Per1*, (**D**) *Per2*, (**E**) *Per2*, (**F**) *Rev-erb alpha*, (**G**) *Npy*, (**H**) *Agrp* (n = 5 per condition). (**I–K**) Induction of *Npy* (**I**), *Agrp* (**J**) and *Per2* (**K**) mRNA levels in *Bmal1*-deficient N44 cells after Adn

*Figure 6 continued on next page*

Figure 6 continued

treatment and in the mediobasal hypothalamus (MBH) after *i.v.* Adn injection in *Bmal1* knockout mice (n = 3 per condition). */**/***: p<0.05/0.01/0.001, unpaired Student's t-test. (**L**) Luciferase promotor assay for *Npy* in HEK293T cells after transient expression of increasing doses of *Clock/Bmal1* or *Cry1* (n = 6–10). */***: p<0.05/0.001 *vs.* -/-, ###: p<0.001 *vs.* 40/-, 1-way ANOVA. (**M**) Expression of *Adipor1/2* mRNA levels after shRNA-mediated knockdown of *Adipor1* (blue) or *Adipor2* (green) in N44 cells (n = 3 per condition). (**N, O**) Expression of *Bmal1* (**N**) and *Npy* (**O**) mRNA after *Adipor1/2* knockdown in N44 cells (n = 3–4 per condition). (**P**) Adiponectin-induced phase shift of N44/*Bmal1-luc* rhythms after *Adipor1/2* knockdown in N44 cells (n = 4 per condition). */**/***: p<0.05/0.01/0.001 *vs.* Con, 1-way ANOVA. (**Q**) Expression of *Adipor1* mRNA in the mediobasal hypothalamus (MBH; n = 7–10) and of ADIPOR1 protein in the arcuate nucleus (ARC; inset; n = 6–9) after shRNA-mediated viral knockdown of *Adipor1*. (**R, S**) Daily (LD; **R**) and circadian (DD; **S**) food intake profiles of mice after shRNA-mediated viral knockdown of *Adipor1* in the MBH (KD) and of scramble-AAV treated control animals (Con) (n = 9–13 per group). (**T**) Relative light phase food intake in LD (left) and rest phase food intake in DD (right) in *Adipor1*-KD animals (KD) and controls (Con) (n = 9–13 per group). */**/***: p<0.05/0.01/0.001 *vs.* Con, unpaired Student's t-test (**Q, T**) or 2-way ANOVA with Sidak's multiple comparisons (**R, S**). All data are averages ± SEM. Box plots show medians, quartiles and min/max. p-values: (**B**) 0.0084 (N44, +/- Adn), 0.0309 (MBH, +/- Adn); (**G**) < 0.001 (N44, +/- Adn and MBH, +/- Adn); (**H**) 0.0002 (N44, +/- Adn), 0.0059 (MBH, +/- Adn); (**L**) 0.019 (10 /- *vs.* -/-),<0.001 (20 /- and 40 /- *vs.* -/-),<0.001 (40/12, 40/25 and 40/50 *vs.* 40/-), ANOVA dF = $F_{(7, 51)}$=15.63; (**M**) < 0.001 (*Adipor1* KD, *Adipor1* and *Adipor2 vs.* control and *Adipor2* KD, *Adipor2*), ANOVA dF = $F_{(4, 10)}$=68.39; (**N**) 0.002 (*Adipor1* KD *vs.* control),<0.001 (*Adipor2* KD and *Adipor1/2* KD *vs.* control), ANOVA dF = $F_{(3, 10)}$ =68.50; (**O**) 0.010 (*Adipor1* KD *vs.* control), 0.005 (*Adipor2* KD *vs.* control), 0.016 (*Adipor1/2* KD *vs.* control), ANOVA dF = $F_{(3, 8)}$=31.95; (**P**) < 0.001 (*Adipor1* KD and *Adipor1/2* KD *vs.* Con), ANOVA dF = $F_{(3, 12)}$=25.72; (**Q**) 0.006 (mRNA) and <0.001 (protein), unpaired Student's t-test; (**R**) 0.037 (KD *vs.* Con, ZT12-18), ANOVA dF = $F_{(3, 54)}$=2.359 (interaction)/37.82 (time)/0.56 (knockdown); (**S**) 0.002 (KD *vs.* Con, ZT12-18), ANOVA dF = $F_{(3, 66)}$=5.337 (interaction)/87.32 (time)/1.43 (knockdown); (**T**) 0.0454 (LD, Con *vs.* KD) and 0.0199 (DD, Con *vs.* KD), unpaired Student's t-test.

The online version of this article includes the following source data and figure supplement(s) for figure 6:

**Source data 1.** Raw data of experiments shown in *Figure 6*.

**Figure supplement 1.** Adiponectin induces *Bmal1* expression in synchronized N44 cells.

---

infusion (*Figure 6B,C*) while total (24 hr) food intake was not affected (*Figure 6D*). Remarkably, mice receiving rhythmic AdipoRon infusions lost about one third of the body weight they had gained during the preceding six weeks of HFD, while vehicle and constant AdipoRon treated mice did not show such response (*Figure 6E*). Consistently, at the end of the experiment eWAT of the rhythmically AdipoRon-treated mice was significantly reduced compared to mice receiving vehicle or the drug at constant levels (*Figure 6F*).

To test if this body weight resetting effect of rhythmic AdipoRon infusion was mediated *via* its function in *Bmal1* regulation, we repeated the experiment in *Bmal1* deficient mice. Unlike in WT animals, AdipoRon treatment had no effect on food intake profiles in the mutants which showed non-rhythmic feeding patterns under all treatment conditions (*Figure 8G*). Likewise, AdipoRon treatment did not affect body weight regulation nor adiposity in Bmal1 deficient animals suggesting that *Bmal1* is necessary to mediate ADIPOQ's effects on diurnal appetite and body weight regulation (*Figure 8H,I*).

## Discussion

In the current study, we describe a circadian adipose-brain circuit signaling peripheral metabolic state to the brain to adjust circadian feeding behavior in mice. The adipose released peptide ADIPOQ acts as a *zeitgeber* of mediobasal hypothalamic circadian clocks and feeding rhythms. ADIPOQ induces expression of the clock gene *Bmal1* in MBH neurons *via* AdipoR1. Both, *Adipoq*-deficient mice and MBH *Adipor1* KD mice exhibit dampened feeding rhythms correlating with altered clock gene responses in the MBH, while retaining normal circadian locomotor activity rhythms. Timed activation of MBH ADIPOQ signaling during the active phase, however, can normalize the abnormal feeding rhythms induced by *Adipoq*-deficiency and improve body weight regulation in diet-induced obese wild-type mice.

This study has several limitations. First, most of the in-vivo data were obtained from male mice only. While the effects of ADIPOQ deficiency on appetite regulation seem preserved in both genders (compare *Figure 2* and *Figure 3* and accompanying supplements), it needs to be shown if pharmacological manipulation of ADIPOQ signaling in the MBH can rescue appetite and body weight regulation also in females. Second, while we provide conclusive evidence that ADIPOQ acts *via* the circadian clock system to control food intake and body weight, we do not specifically show that this requires resetting of clocks in the MBH. Thus, indirect input involving clocks in, for example other brain regions cannot be excluded. Finally, while rhythmic activation of ADIPOQ signaling in the

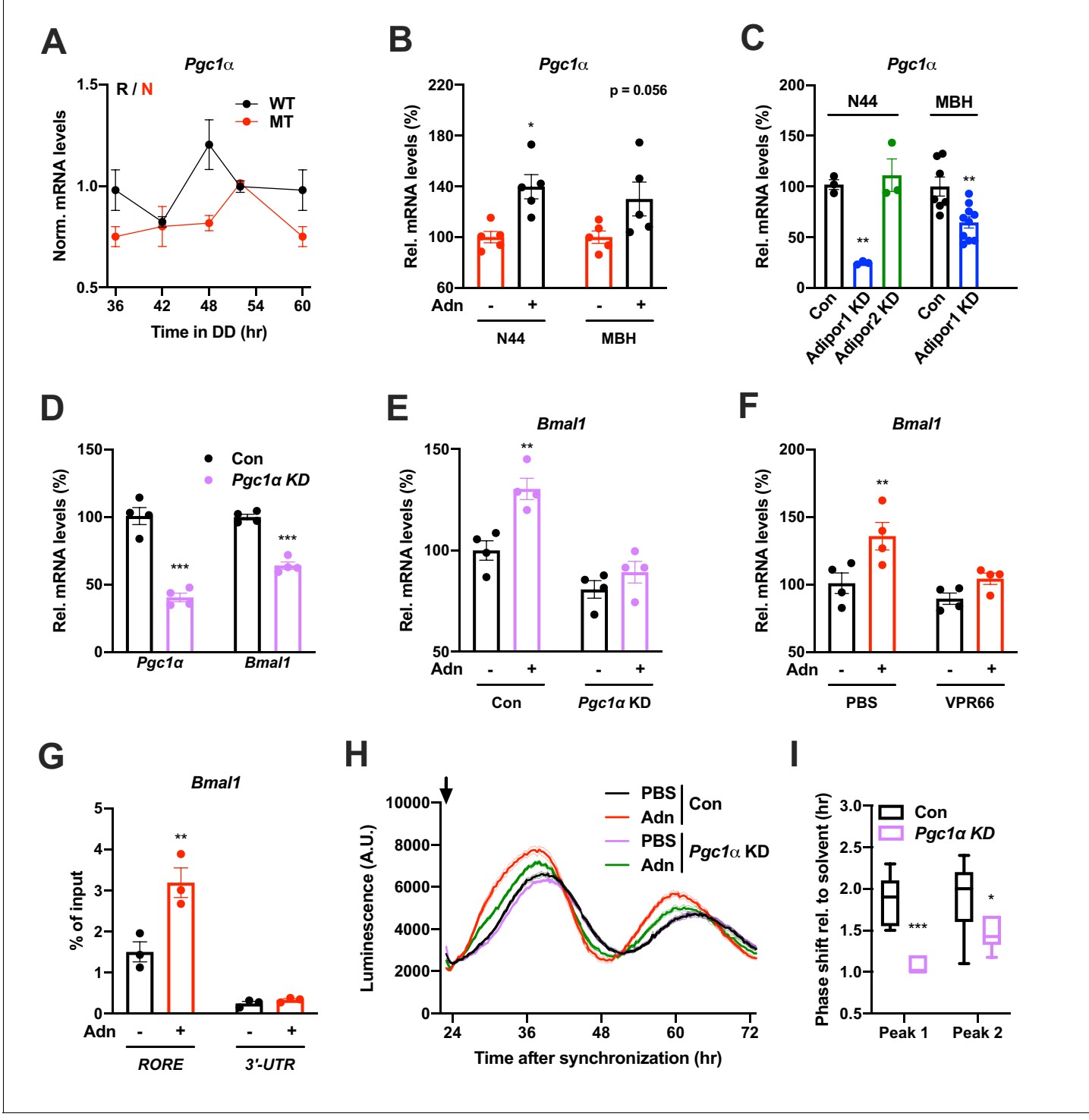

**Figure 7.** PGC1a-mediated induction of *Bmal1* by ADIPOQ. (**A**) Circadian profile of *Pgc1a* mRNA levels in the MBH of WT (black) and *Adipoq* deficient (MT) mice (red; n = 3 per time point). (**B**) Induction of *Pgc1a* mRNA levels in N44 cells after Adn treatment and in the mediobasal hypothalamus (MBH) after *i.v.* Adn injection in wild-type mice (n = 5). (**C**) Expression of *Pgc1a* in N44 cells and in the MBH after shRNA-mediated knockdown of *Adipor1* or *Adipor2* (n = 3 for N44 cells and 7 (Con)/10 (KD) in MBH). (**D**) mRNA levels of *Pgc1a* (left) and *Bmal1* (right) in non-synchronized N44 cells after shRNA-mediated knockdown of *Pgc1a* (n = 4). (**E**) *Bmal1* mRNA levels in unsynchronized N44 cells in response to Adn with (right) or without (left) *prior* knockdown of *Pgc1a* (n = 4). (**F**) *Bmal1* mRNA levels in unsynchronized N44 cells in response to Adn with (right) or without (left) *prior* treatment with 5 µM VPR66 (n = 4). (**G**) ChIP-qPCR for PGC1a binding to *Bmal1 RORE* (and *Bmal1 3'-UTR* as negative control) in unsynchronized N44 cells in response to Adn treatment (n = 3). (**H, I**) Response of *Bmal1-luc* luminescence rhythms in synchronized N44 cells to Adn treatment at 23 hr after synchronization

*Figure 7 continued on next page*

*Figure 7 continued*

(arrow) with or without *prior* knockdown of *Pgc1a*. (H) Normalized luminescence data (n = 3 per condition). (I) Phasing of the first and second peak after Adn treatment relative to solvent (n = 8). All data are means ± SEM. Box plots show medians, quartiles and min/max. p-values: (B) 0.0116 (N44, +Adn *vs.* -Adn), ANOVA dF = $F_{(1, 16)}$=0.299 (interaction)/0.294 (tissue)/15.60 (treatment); (C) 0.0001 (N44, *Adipor1* KD *vs.* Con), 0.0020 (MBH, *Adipor1* KD *vs.* Con), ANOVA dF = $F_{(1, 19)}$=0.028 (interaction)/0.044 (tissue)/40.90 (treatment); (D) < 0.0001 (*Pgc1a*, *Pgc1a* KD *vs.* Con),<0.0001 (*Bmal1*, *Pgc1a* KD *vs.* Con), ANOVA dF = $F_{(1, 12)}$=9.693 (interaction)/8.566 (gene)/151.5 (treatment). (E) 0.0019 (Con, +Adn *vs.* -Adn), ANOVA dF = $F_{(1, 12)}$=4.894 (interaction)/37.05 (genotype)/15.39 (treatment); (F) 0.0087 (PBS, +Adn *vs.* -Adn), ANOVA dF = $F_{(1, 12)}$=2.044 (interaction)/9.281 (inhibition)/12.45 (treatment); (G) 0.0013 (*RORE*, +Adn *vs.* -Adn), ANOVA dF = $F_{(1, 8)}$=13.17 (interaction)/87.25 (site)/16.20 (treatment); (I) < 0.0001 (peak 1, Con *vs.* KD), 0.0258 (peak 2, Con *vs.* KD), ANOVA dF = $F_{(1, 24)}$=2.809 (interaction)/3.192 (peak)/29.93 (genotype).

The online version of this article includes the following source data and figure supplement(s) for figure 7:

**Source data 1.** Raw data of experiments shown in *Figure 7*.
**Figure supplement 1.** Knockdown of *Pgc1a* affects circadian rhythms in N44 cells.

hypothalamus exerts short-term weight regulating effects under obesogenic conditions, it remains to be shown to which extent this also applies to more chronic conditions and different types of diets.

Established circadian clock-resetting mechanisms involve acute induction of *Per* genes after activation of cAMP- (e.g. photic resetting in the SCN) or glucocorticoid-dependent (e.g. in the liver) pathways (*Dibner et al., 2010*). The ADIPOQ-induced phase resetting mechanism described here is special with respect to its distinct phase-response profile and its ability to induce *Bmal1* instead of *Per* expression. Our data suggest that ADIPOQ-ADIPOR1 signaling plays a crucial role in regulating the circadian effects of ADIPOQ in the MBH. Interestingly, while *AdipoR1* is quite widely expressed (*Thundyil et al., 2012*), we observed a marked tissue specificity of the circadian effects of ADIPOQ. As one option, the antagonistic effect of ADIPOR2 on *Bmal1* expression that we observed may counterbalance ADIPOR1's effects in some cell types. The mechanism and physiological relevance of ADIPOR2's circadian effects warrant further studies. Also, expression profiles of both ADIPOQ receptors show anti-phasic regulation and Adipor1 mRNA levels do not align with *Bmal1* and *Npy* expression in the MBH. This suggests that ADIPOR1 levels are not rate-limiting for ADIPOQ's effects. Alternatively, both receptors may interact in mediating clock and appetite regulating effects. ADIPOQ is known to activate multiple signaling pathways some of which have previously been implicated in clock regulation (*Kubota et al., 2007*). Thus, further context-dependent mechanisms modifying the effects of ADIPOQ on clock function may exist.

In line with previous studies, we observed that transcription of appetite-regulating neuropeptides in the MBH is subject to regulation by the circadian clock (*Arble et al., 2009*; *Cedernaes et al., 2019a*). In particular, we provide evidence that expression of *Npy* and *Agrp* is regulated by the MBH clock. Our data support a model by which dampened MBH clock gene rhythms in *Adipoq* deficient mice may cause the blunted circadian rhythms of orexigenic neuropeptide expression and feeding behavior. It has been shown that central ADIPOQ application has an acute appetite-promoting effect *via* AdipoR1-AMPK activation in the MBH (*Iwabu et al., 2010*) suggesting a likely downstream target for mediating the clock-regulating effects of ADIPOQ.

Of note, ADIPOQ has pleiotropic effects in regulating energy homeostasis. For example, it has recently been shown to differentially regulate the electrophysiological responses of AgRP/NPY and POMC neurons *via* a phosphatidylinositol 3-kinase (PI3K) dependent pathway (*Suyama et al., 2017*). How this and other signaling pathways engaged by ADIPOQ interact with the clock machinery is currently unknown. It had previously been observed that *Adipoq*$^{-/-}$ mice were hypophagic and gained less weight under HFD, though – in line with our observations – food intake and bodyweight were unaltered under normal chow conditions (*Kubota et al., 2007*).

Given the rhythmic abundance and metabolic regulation of circulating ADIPOQ, it is conceivable that the altered feeding rhythm observed in *Adipoq*-deficient mice under *ad-libitum* conditions may derive from alterations in clock-dependent and -independent pathways. Likewise, the clock machinery also has diverse effects on neuronal functions such as neuronal excitability and neurotransmitter secretion (*Allen et al., 2017*). It has recently been shown that clocks in NPY/AgRP neurons regulate circadian food intake rhythms (*Cedernaes et al., 2019a*). Thus, NPY/AgRPergic clocks in the MBH may mediate ADIPOQ's effects on circadian feeding behavior.

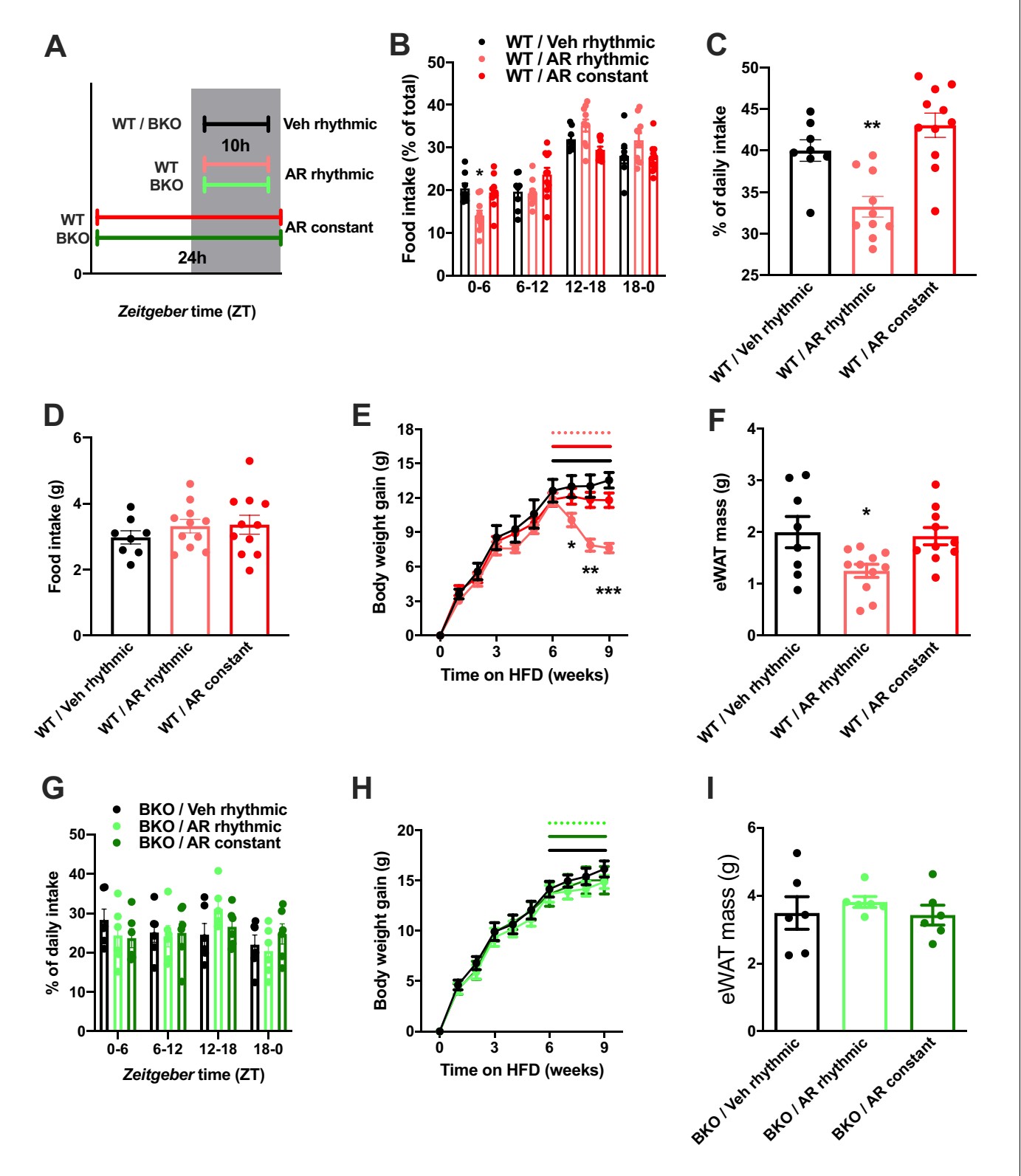

**Figure 8.** Rhythmic AdipoRon administration rescues food intake rhythms and body weight in obese male mice. (**A**) Treatment regimen and groups. (**B**–**D**) Daily food intake profiles (**B**), relative light phase food intake (**C**) and total daily food intake (**D**) of HFD-fed male WT mice treated 10 hr or 24 hr per day *i.c.v.* with AdipoRon or vehicle. (**E**) Body weight development of HFD-fed WT mice before and during constant or rhythmic *i.c.v.* AdipoRon treatment (lines indicate treatment phases). (**F**) Epididymal adipose tissue (eWAT) mass of HFD-fed WT mice after 3 weeks of rhythmic or constant *i.c.v.*

*Figure 8 continued on next page*

Figure 8 continued

treatment with AdipoRon (A-F: n = 8–11 per group). (**G**) Daily food intake profiles of HFD-fed male *Bmal1* deficient (BKO) mice treated 10 hr or 24 hr per day *i.c.v.* with AdipoRon or vehicle. (**H**) Body weight development of HFD-fed BKO mice before and during constant or rhythmic *i.c.v.* AdipoRon treatment (n = 6 per group; lines indicate treatment phases). (**I**) eWAT mass of HFD-fed BKO mice after 3 weeks of rhythmic or constant *i.c.v.* treatment with AdipoRon (G-I: n = 6 per group). Shown are averages ± SEM. */**/***: p<0.05/0.01/0.001, 1-way ANOVA (**C, D, F**) or 2-way ANOVA with Sidak's multiple comparisons (**B, E**). p-values: (**B**) 0.002 (WT/AR rhythmic *vs.* WT/Veh rhythmic; ZT0-6), ANOVA dF = $F_{(6, 78)}$=4.933 (interaction)/ (61.05 (time)/ 0.03 (treatment); (**C**) 0.007 (WT/AR rhythmic *vs.* WT/Veh rhythmic), ANOVA dF = $F_{(2, 26)}$=14.47; (**E**) 0.04 (WT/AR rhythmic *vs.* WT/Veh rhythmic; week 7), 0.001 (WT/AR rhythmic *vs.* WT/Veh rhythmic; week 8),<0.001 (WT/AR rhythmic *vs.* WT/Veh rhythmic; week 9), ANOVA dF = $F_{(18, 234)}$=13.51 (interaction)/550.1 (time)/3.68 (treatment); (**F**) 0.033 (WT/AR rhythmic *vs.* WT/Veh rhythmic), ANOVA dF = $F_{(2, 26)}$=4.709.

The online version of this article includes the following source data for figure 8:

**Source data 1.** Raw data of experiments shown in *Figure 8*.

*Adipoq* mRNA levels in adipose tissues are subject to circadian clock regulation (*Barnea et al., 2015*) and ectopic ADIPOQ overexpression can alter the response of the circadian system to metabolic challenges in mice (*Hashinaga et al., 2013*). These data, together with (i) our observations about the clock-resetting properties of ADIPOQ, (ii) the metabolic state-dependent regulation of ADIPOQ in the circulation and (iii) its receptors and downstream targets in the hypothalamus, prompted us to speculate that ADIPOQ signaling may be involved in the food-dependent regulation of MBH clocks. In line with this, we found that MBH clocks of *Adipoq* KO mice displayed dampened food intake rhythms that can be restored by rhythmic activation of central ADIPOQ signaling. Hypothalamic NPs such as NPY and AgRP are regulators of food intake, circadian appetite rhythms and metabolic appetite regulation (*Cedernaes et al., 2019a*; *Cedernaes et al., 2019b*). Furthermore, previous studies indicate that hypothalamic extra-SCN clocks affect food intake (*Bechtold and Loudon, 2013*; *Cedernaes et al., 2019b*). Given its role in regulating the expression of orexigenic NPs, we reasoned that ADIPOQ may exert its effects on feeding rhythms through resetting of clock-controlled NP expression in the MBH.

In humans, increased eating at night time is associated with metabolic disorders and obesity (*Stunkard, 2011*). Likewise, mice subjected to temporal food restriction during the day – their normal inactive phase – also gain significantly more weight compared to their night-fed counterparts (*Arble et al., 2009*). Intriguingly, as it had also been shown in the current study, mice fed on HFD spontaneously show dampened circadian feeding rhythms (*Kohsaka et al., 2007*). On the contrary, protocols restricting feeding to the active phase are protective against diet-induced obesity and associated metabolic disorders in mice and humans independent of total energy intake (*Chaix et al., 2014*; *Gill and Panda, 2015*; *Hatori et al., 2012*). These observations clearly demonstrate the physiological and clinical relevance of robust daily energy intake rhythms, which are often compromised in metabolically challenged individuals, resulting in a vicious cycle. Impaired leptin function has recently been implicated in this process (*Ando et al., 2011*), but our data indicate that leptin may not directly act on hypothalamic clocks (*Grosbellet et al., 2015*). We here show that the obesogenic and appetite rhythm-blunting effects of HFD can be counteracted by rhythmic activation of ADIPOQ signaling in the brain, proposing that ADIPOQ may act as a peripheral metabolic feedback signal to the molecular clockwork in MBH neurons. In line with this notion, constant infusion of ADIPOQ had no effect on food intake patterns, body weight and adiposity neither in WT nor in clock deficient (BKO) mice. Temporal variation of ADIPOQ signaling appears necessary to mediate this effect in WT mice, while BKO mice are resistant to ADIPOQ treatment. Together, these data suggest that rhythmic activation of hypothalamic ADIPOQ signaling may reset blunted MBH clock function, thereby restoring appetite rhythms and body weigh regulation. In line with this, it has previously been shown that transcriptional control by clocks in AgRP/NPY neurons aligns hunger and food acquisition with circadian behavior (*Cedernaes et al., 2019a*).

As such ADIPOQ would form an adipose-brain circuit regulating circadian behaviors in response to variations in metabolic state. A similar feedback has been suggested in another study: deletion of clock function specifically in adipocytes results in similarly blunted diurnal feeding rhythms in mutant mice together with lower levels of poly-unsaturated fatty acids (PUFAs) in the blood (*Paschos et al., 2012*). Moreover, supplementing PUFAs with the diet can restore normal feeding rhythms and expression of an-/orexigenic MBH genes in the adipose clock mutants (*Paschos et al., 2012*). ADIPOQ levels were unaltered in these mutants while we did not investigate PUFA levels in our study.

Also, MBH clock function has not been assessed in the adipose clock mutants. ADIPOQ- and PUFA-mediated pathways converge on an-/orexigenic gene expression in the MBH and appetite regulation. Both independently seem to be necessary for normal diurnal food intake rhythms and it will be interesting to study their interaction under different diet conditions.

In humans, blunted diurnal variations ADIPOQ in the blood have been observed in obese individuals which can be restored by weight loss induced by surgical gastric bypass (*Calvani et al., 2004*; *Yildiz et al., 2004*). Only rhythmic - but not constant - pharmacological activation of central ADIPOQ signaling was able to normalize the dampened diurnal feeding rhythms in *Adipoq*-deficient and HFD-treated mice, emphasizing the circadian role of central ADIPOQ signaling. Of note, ADIPOQ can exist in different forms in the circulation with different molecular properties. Among these, the high-molecular weight oligomers have been shown to be the most potent and clinically relevant form in the body (*van Andel et al., 2018*). It will be interesting to test how the circadian clock affects ADIPOQ processing.

In conclusion, our data suggest blunted ADIPOQ signaling in obesity, may – besides its established impact on insulin signaling (*Ruan and Dong, 2016*) – further deteriorate metabolic function *via* circadian disruption of central metabolic control circuits. *Vice versa*, restoring central ADIPOQ signaling rhythms may be an attractive target to counteract the obesogenic effects of high-calorie diets and a countermeasure against further weight gain in obese patients.

# Materials and methods

## Animals and circadian behavioral experiments

Wild-type (WT), *Adiponectin* (MT), *Bmal1* knockout (BKO) (*B6;129-Adipoq^{tm1Chan/J}*; JAX stock #008195 and B6.129-*Arntl^{tm1Bra}*/J; JAX stock #009100) and *PER2::LUCIFERASE* (PER2::LUC) mice (*B6.129S6-Per2^{tm1Jt/J}*; JAX stock #006852) were purchased from the Jackson's Laboratory (ME, USA). They were maintained at the institutes' animal facilities in Göttingen or Lübeck. All mice used were kept on a C57BL6J genetic background. For all experiments, unless stated otherwise, male mice were individually housed under 12 hr light, 12 hr dark conditions (LD;~100 lux in the light phase) with *ad-libitum* access to chow pellets (normal chow (15% kJ fat, Altromin 1320) or high-fat diet (60 % kJ fat, Altromin c1090-60) and water. For experiments in constant darkness (DD), mice were first entrained under LD conditions for at least one week and then released into DD for the indicated timespans. Mice in darkness conditions were handled under dim red light (including the *i. v.* injection experiments at ZT 21). Behavioral experiments were performed on animals aged 10–12 weeks at the beginning of the experiment. Circadian parameters of locomotor activity rhythms were assessed by running-wheels and analyzed with Clocklab Analysis software (Actimetrics, IL, USA). Molecular analyses were performed on 16–24 week-old animals. In all experiments involving organ harvesting, mice were sacrificed by cervical dislocation. To isolate the MBH and the SCN for molecular analyses, 1 mm thick slices containing the respective regions were isolated with the aid of a rodent brain matrix (ASI Instruments, MI, USA) and were then further dissected under a microscope. All animal experiments were done after ethical assessment by the institutional animal welfare committee and licensed by the Office of Consumer Protection and Food Safety of the State of Lower Saxony or the Ministry of Agriculture of the State of Schleswig-Holstein in accordance with the German Law of Animal Welfare (TierSchG). Sample size estimation was done for behavioral animal experiments. Parameter estimates were: type: repeated measures ANOVA – within/between interaction; effect size 0.2; alpha 0.05; beta 0.8; two groups; four measurements (time points); repeated measures correlation 0.4; epsilon 1. This yielded a total sample size of 44 = 4 replicates per time point. Following the 3R principle, smaller sample sizes were used when effect sizes were larger than expected. Experiments were repeated where larger sample sizes are indicated, and data pooled.

## Food restriction paradigm

Mice were adapted to a 4 hr feeding regime by reducing access to food in four steps starting with *ad-libitum* access to food for 12 hr, which was reduced to 9 hr food access the following day (either ZT3-12 or ZT15-24). On day 3 and 4 food access was further shortened by 2 hr per day. The restricted 4 hr food access (ZT7-11 or ZT19-23) occurred for two weeks.

Cell culture and circadian luminescence recording mHypoE-N44 (Cedarlane Labs, NC, USA) and HEK293T cells (ATCC, Wessel, DE) – authenticated by STR profiling and tested negative for mycoplasms by PCR – were maintained in DMEM with 2 mM stable glutamine supplemented with 10% fetal bovine serum (FBS) and 10,000 U penicillin/streptomycin at 37°C with 5% $CO_2$. Cells stably expressing *Bmal1-luc* reporter *via* lentiviral transduction were generated polyclonally by puromycin selection. For circadian luminescence measurements, confluent cells seeded in 96-well plates were synchronized by 100 nM dexamethasone treatment for 2 hr. After that, medium was replaced with recording medium (DMEM without phenol red supplemented with 2 mM stable glutamine, 3 mM sodium carbonate, 10 mM HEPES, 2% B-27 supplement, 10,000 U penicillin/streptomycin and 0.5 mM D-luciferin). Plates were sealed with transparent films and luminescence was recorded at 34°C using the Berthold TriStar LB 941 plate reader (Berthold Technologies, Wildbach, DE). *Bmal1* KO N44 cells were generated using CRISPR (clustered regularly interspaced short palindromic repeats) based mutagenesis. Cells were transduced with a lentivirus expressing *SpCas9* (from *Streptococcus pyogenes*), a single-guide (sg)RNA cassette targeting the N-terminal DNA-binding bHLH (basic helix-loop-helix) domain of mouse *Bmal1* and puromycin N-acetyltransferase. Puromycin-resistant clones were picked and expanded for validations. Clones with successful *Bmal1* KO were confirmed with the loss of BMAL1 expression with Western blot (details as below; the BMAL1 antibody used is C-terminal specific) and the loss of cellular circadian rhythm using *Bmal1-luc* circadian luminescence assays.

## Primary hypothalamic neuronal culture

Hypothalami of E16 embryos were dissected and isolated using the papain dissociation system (Worthington Biochemical, NJ, USA) according to the manufacturer's protocol. $3.25 \times 10^5/cm^2$ viable cells in plating medium (neural basal medium supplemented with 2 mM stable glutamine, 2% B-27, 10% FBS and 10,000 U penicillin/streptomycin) were seeded onto 96-well plates double-coated with poly-D-lysine and laminin. On the next day, the plating medium was replaced with feeding medium (same as plating medium, but without FBS) and transduced with Bmal1-luc lentivirus. 24 hr later, half the volume of the old medium was refreshed with fresh feeding medium containing 5 μM cytosine arabinoside. Half the volume of the old medium was subsequently refreshed every 3 days. On day in vitro (DIV) 9, neurons were synchronized with 100 nM dexamethasone treatment for 2 hr and subjected to bioluminescence experiments in feeding medium supplemented with 0.5 mM D-luciferin.

## ARC/ME slice cultures

Luminescence was measured from cultured ARC/ME slices of heterozygous *PER2::LUC* mice as described previously (*Yoo et al., 2004*) and *Guilding et al., 2009*). Briefly, brains were isolated and harvested in ice-cold Hank's balanced salt solution (HBSS). 300 μm ARC/ME containing coronal slices (−1.80 mm and −2.10 mm relative to bregma) were prepared using a vibratome (Thermo Scientific, MA, USA). Bilateral ARC/ME regions were further dissected under a dissecting microscope and then immediately placed onto a culture plate insert (Merck Millipore) in 35 mm Petri dishes filled with 1 ml recording medium (same as the one used for cell culture). Luminescence was measured in a Lumi-Cycle (Actimetrics) at 32.5°C.

## Substances used for treatments in cells and animals

Forskolin (Fors), dexamethasone (Dex), globular adiponectin (Adn), and AdipoRon were purchased from Phoenix Pharmaceuticals (Karlsruhe, DE); leptin and full-length adiponectin were purchased from Enzo Life Science (Loerrach, DE); ghrelin, glucagon, and nesfatin were purchased from Bachem (Bubendorf, CH); resistin was purchased from Prospec (NJ, USA) and visfatin from Aviscera Bioscience (CA, USA); VPR66 was purchased from BioMol (Hamburg, DE).

Plasmid constructs pLKO-GFP-WPRE vector was used for all shRNA knockdown experiments in cell culture, a modified version of the original pLKO.1-TRC backbone (Addgene plasmid #10878; *Moffat et al., 2006*) in which the puromycin resistance open reading frame (ORF) was replaced with a GFP-WPRE sequence from pLenti-CMV-GFP-Zeo (637-7) (Addgene plasmid #17449; *Campeau et al., 2009*) with *Kpn I* and *BamH I* restriction sites. Targeting nucleotide sequences of shRNA used were selected from the RNAi Consortium (TRC) database (Broad Institute, MA, USA) and cloned into pLKO-GFP-WPRE using *Age I* and *EcoR I* restriction sites. The same scramble and

*Adipor1* KD sequences were also used in the AAV in vivo knockdown experiments (see below). *Npy-luc* plasmid was generated by cloning a ~ 2 kb 5'-promoter fragment of the murine *Npy* gene (GenBank accession #NC_000072.6; *Fick et al., 2010*) into *pGL4* vector (Promega, Mannheim, DE) with *Kpn I* and *Bgl II* restriction sites. This fragment has previously been identified to contain multiple *E-boxes* (*Fick et al., 2010*). To generate the *Bmal1* KO lentiviral plasmid, a guide RNA targeting the bHLH domain locating on the exon 5 of mouse *Bmal1* was cloned into *lentiCRISPR v2* (Addgene plasmid #52961) (*Sanjana et al., 2014*) with *BsmBI* restriction site. This guide RNA was designed with the aid of the GPP sgRNA Designer (Broad Institute). The integrity of all recombinant plasmids was confirmed by sequencing.

## Lentivirus production and transduction

The *Bmal1-luc* encoding *pBluF-puro* plasmid was a kind gift from Prof. Steven Brown (University of Zurich, CH). To produce lentiviral particles, HEK293T cells were co-transfected with *psPAX2* (Addgene plasmid #12260), *pMD2.G* (Addgene plasmid #12259, Prof. Didier Trono, EPFL, CH), and *pBluF-puro* or *pLKO-WPRE-GFP* carrying the respective shRNA sequences or *lentiCRISPR v2* expressing SpCas9 and *Bmal1* sgRNA using Xfect transfection reagent (Clontech, Saint-Germain-en-Laye, FR) according to the manufacturer's protocol. Virus-containing media were pooled and concentrated using LentiX concentrator reagent (Clontech) according to the manufacturer's protocol. Virus titers were determined by quantifying viral genome abundance with qPCR analysis using a primer pair amplifying the *WPRE* element region (forward: 5'-GGCACTGACAATTCCGTGGT-3'; reverse: 5'-AGGGACGTAGCAGAAGGACG-3'). Cells at 50% confluence were transduced with ~1 × 10$^8$ infection units (IFUs) per 1 ml medium in the presence of 8 µg/mL polybrene. Experiments on cells or selection for stably expressing cells were carried out at 72 hr after transduction.

## Quantitative real-time polymerase chain reaction (qPCR)

Isolated tissues or cells were harvested and kept in RNAlater solution (Life Technologies) according to the manufacturer's protocol. Total RNA of tissues and cell cultures was extracted using either TRIzol reagent (Life Technologies) or RNeasy FFPE Kit (Qiagen, Hilden, DE) (in-vivo Adipor1 knockdown) according the manufacturer's protocol. cDNA synthesis was performed using the High-Capacity cDNA Reverse Transcription Kit (Life Technologies) with random hexamer primers. qPCR was performed using GoTaq qPCR Master Mix (Promega) on a CFX96 thermocycler (Bio-Rad, Munich, Germany). Relative gene expression was quantified using the ∆∆ threshold cycle (Ct) method with adjustments to the amplification efficiencies of individual primer pairs. *Eef1α* was used as the reference gene for all experiments. Primer sequences are shown in the list of oligos below. No sample size calculation was done for gene/protein expression etc. Standard replicate number was 3. Experiments were repeated where larger sample sizes are indicated, and data pooled.

## Chromatin immunoprecipitation (ChIP)

2 hr after ADN treatment, cells were fixed with 1% formaldehyde at 37°C for 10 min. Cross-linking was stopped by adding 1/20 vol of 2.5 M glycine and incubation at room temperature for 5 min. Cells were lysed with SDS lysis buffer (1% SDS, 50 mM EDTA, 50 mM Tris-HCl pH 8.1 with 1x cOmplete protease inhibitor cocktail) and subjected to sonication using Branson 450 sonicator (amplitude: 50, duty: 30%, duration: 30 s, 6 repeats with 1 min intervals between each sonication) which yielded majority of DNA fragments with size 200–1000 bp. Precleared samples were incubated overnight at 4°C with anti-PGC1α antibody (H300; Cat# sc-13067, Santa Cruz, US). The samples were incubated with Protein G agarose beads (Thermo Scientific) for 1 hr at 4°C followed by intensive washings: 1x with low salt washing buffer (0.1% SDS, 1% Triton, 2 mM EDTA, 20 mM Tris-HCl, pH8.1, 150 mM NaCl), 1x with high salt washing buffer (0.1% SDS, 1% Triton, 2 mM EDTA, 20 mM Tris-HCl, pH8.1, 500 mM NaCl), 1x with LiCl washing buffer (0.25M LiCl, 1%NP40, 1% Na deoxycholate,1mM EDTA, 1 mM Tris-HCl, pH8.1) followed by 2x TE buffer (10 mM Tris-HCl pH 8.0, 1 mM EDTA). Afterward, immune complexes were eluted with an elution buffer (1% SDS, 0.1M NaHCO$_3$). The eluates were then treated with Proteinase K and further subjected to phenol-chloroform-isoamyl alcohol purification. Purified samples were then subject to qPCR analysis with primer pairs flanking the RORE and the 3'UTR of *Bmal1* gene and values were normalized to percentage of pre-IP input. Primer sequences are listed in the List of Oligonucleotides Table.

## Western blot

To detect BMAL1 protein in N44 cells, cells were lysed with lysis buffer (1% triton X-100, 2% sodium dodecylsulfate (SDS), 1% sodium deoxycholate, 1 % NP40 and 1x cOmplete protease inhibitor cocktail (Roche, Grenzach-Wyhlen, DE) in Tris-buffered saline (TBS) and subjected to sonication using a Branson 450 sonifier (Thermo Scientific; amplitude: 50, duty: 30%, duration: 30 s) and subsequent Western blot analysis according to standard protocols of SDS-polyacrylamide gel electrophoresis (PAGE) and Western blot using anti-BMAL1 antibody (dilution 1:1,000; Novus Biologicals, Cambridge, UK). Protein concentrations were determined by BCA protein assay kit (Thermo Scientific); endogenous β-tubulin levels determined with anti-β-tubulin (dilution 1:1000; Cell Signaling Technology) were used as loading reference. Densitometric analysis of band intensity was performed with Quantity One software (Bio-Rad).

## ELISA analysis of plasma adiponectin

Trunk blood was collected from decapitated animals into EDTA-containing tubes (Sarstedt, Nümbrecht, DE). Plasma concentrations of adiponectin were determined with an ELISA detection kit for mouse adiponectin (Adipogen, CA, USA) according to the manufacturer's protocol using the Epoch microplate spectrophotometer (Biotek Instruments, VT, USA).

## Adeno-associated virus (AAV) production

The AAV-DJ system was used to produce AAV particles of recombinant DJ serotype carrying the shRNA-GFP expression cassette according to the manufacturer's protocol (Cell Biolabs, CA, USA) (*Grimm et al., 2008*). Briefly, the scramble and *Adipor1* KD1 shRNA sequences were cloned into the *BamH I* and *EcoR I* sites of pAAV-U6-GFP expression vector. The resultant plasmids were co-transfected with *pAAV-helper* and *pAAV-DJ* plasmids to HEK293T cells using the calcium phosphate precipitation method according to the manufacturer's protocol. 60 hr after transfection, cells were lysed in TBS containing 1% triton X-100 with three consecutive freeze-thaw cycles. After clarification, the cell lysate was reunited with culture medium and treated with 1000 units of benzonuclease for 1 hr at 37°C. The mixture was then subjected to filtration with a 0.2 μm low-protein binding filter. The filtrate was loaded to AVB HiTrap sepharose columns (GE healthcare) (*Smith et al., 2009*) with a perfusion pump at a flow rate of 0.75 ml/min. After binding AAV particles were eluted with a low pH elution buffer (0.2 M glycine-HCl, pH 2.7) and immediately neutralized with neutralization buffer (1 M Tris-HCl, pH 8.0). The AAV particles containing eluate was then buffer-exchanged to PBS and concentrated with Amicon Ultra centrifugal filter units with a nominal molecular weight cut-off of 10 kDa (Millipore). AAV particles were retrieved by resuspending the retained solution and then filter-sterilized with a 0.2 μm low-protein binding filter. Viral titers were determined with a commercially available qPCR-based titration kit (Clontech).

## Stereotaxic injections and automated brain infusion

*For in-vivo* knockdown experiments, parenchymal injections were performed under isoflurane anesthesia (CP-Pharma, Burgdorf, DE) and carprofen analgesia (5 mg/kg; Rimadyl, Pfizer, Berlin, DE) in mice with aged 8–10 weeks. 0.5 μl of AAV viral preparation with >$10^{11}$ viral genome copies/ml were bilaterally injected into the ARC (coordinates ±0.3 mm lateral, 1.5 mm posterior and 6.1 mm ventral relative to Bregma) with a 0.5 μl Hamilton glass syringe over 2 min. The injection needle was held in place for another 5 min to prevent backflow of virus solution. Animals were allowed to recover for 3 weeks before further experiments. To validate *Adipor1* knockdown in the ARC, postmortem qPCR analysis and immunohistochemistry were used. For qPCR analysis the ARC was isolated by laser capture microdissection (PALM Microbeam System, Carl Zeiss, Oberkochen, DE) from cryosections (three pooled consecutive sections with a total thickness of 32 μm each). Only data from animals with a marked reduction of *Adipor1* mRNA were included in the final analysis. AdipoR1 protein was investigated by immunohistochemistry. PFA-fixed 12 μm cryo-sectioned brain sections were investigated by using anti-AdipoR1 antibody (dilution 1:20; Abcam ab70362) followed by biotinylated secondary antibody (1:600) detection as described previously (*Koch et al., 2010*).

For timed *i.c.v.* infusions, cannulae (Brain Infusion Kit; ALZET, CA, USA) were unilaterally implanted into the lateral ventricle (0.9 mm lateral and 0.1 mm posterior to Bregma, 2.6 mm ventral to the surface of the skull) of 8-week-old *Adipoq*$^{-/-}$ or WT mice. Cannulae were fixed with superglue

on the skull surface and connected with a subcutaneous tube to programmable iPRECIO SMP-300 pumps (Primetech Corporation, Tokyo, JP) placed under the skin of the back. Skin sutures were further protected with tissue adhesive (Surgibond, Sutures Ltd, Waxham, UK). After 2 weeks of recovery (MT experiment; *Figure 4*) or after 6 weeks of HFD (WT HFD experiment; *Figure 6*), infusion of AdipoRon (1 mg/ml) or vehicle (DMSO/aCSF/PEG-400) was started, either cyclically with infusions starting at ZT13 for 10 hr followed by 14 hr of inactivity or constantly with a maximal flow of 0.5 µl/hr (for treatment regimens see *Figures 4A* and *8A*). Under short isoflurane anesthesia substance solutions in the pumps were refreshed every 7–8 days at ZT4-6. 3–4 days after the last refreshing either food intake data were collected or animals were sacrificed for molecular analyses.

## Statistical analysis

Circadian parameters of luminescence recordings were analyzed using the LumiCycle analysis software (Actimetrics). Briefly, all traces were smoothened by a 5-point moving average and then subjected to baseline subtraction of the 24 hr moving averages and further fitted to a dampened sine wave. Phase shifts were determined by either comparing the fit peak times of substance and control treated cells (all experiments on N44 cells) or comparing the actual peak times after treatment with the extrapolated peak times from pre-treatment sine fits of at least two consecutive circadian cycles before the treatments (all experiments on primary hypothalamic neurons and ARC/ME slices). To construct treatment phase response profiles, the intersection of the ascending cross-section of the sine wave with the x-axis was defined as 0 or $2\pi$ in radian. Dampening rate constants of peak magnitudes were calculated by fitting a one-phase decay equation (Y = Y0 * exp (-K * X) + baseline; where K is the dampening rate constant) to the absolute peak magnitudes of the luminescence rhythms, assuming that the dampening follows an exponential decay pattern. Circadian or diurnal rhythmicity was assessed using CircWave (*Oster et al., 2006*). General statistical analysis was performed using GraphPad Prism 8.0 (GraphPad, CA, USA). Two-tailed unpaired Student's t tests were used for pairwise comparisons. Multi-group analyses were performed with 1-way ANOVA and Tukey's multiple comparisons. Temporal profiling experiments were analyzed by 2-way ANOVAs with Sidak's post-test for comparison of specific time points.

## List of oligonucleotides

**shRNA**

| Target | Sense | References |
|---|---|---|
| scramble | CCTAAGGTTAAGTCGCCCTCG | Addgene plasmid #8455 |
| *Adipor1* KD | GTACGTCCAGGCTTCAAATAA | TRCN0000249147 |
| *Adipor2* KD | GCAGGAATTTCGTTTCATGAT | TRCN0000175771 |
| *Pgc1α* KD | TAACTATGCAGACCTAGATAC | TRCN0000219080 |

**qPCR**

| Target | Forward | Reverse |
|---|---|---|
| *Eef1α* | CACATCCCAGGCTGACTGT | TCGGTGGAATCCATTTTGTT |
| *Bmal1* | TGACCCTCATGGAAGGTTAGAA | CAGCCATCCTTAGCACGGT |
| *Per2* | GCCAAGTTTGTGGAGTTCCTG | CTTGCACCTTGACCAGGTAGG |
| *Dbp* | AATGACCTTTGAACCTGATCCCGCT | GCTCCAGTACTTCTCATCCTTCTGT |
| *Rev-erbα* | AGCTCAACTCCCTGGCACTTAC | CTTCTCGGAATGCATGTTGTTC |
| *Npy* | CTCCGCTCTGCGACACTAC | GGAAGGGTCTTCAAGCCTTGT |
| *Agrp* | ATGCTGACTGCAATGTTGCTG | CAGACTTAGACCTGGGAACTCT |
| *Pomc* | ATGCCGAGATTCTGCTACAGT | TCCAGCGAGAGGTCGAGTTT |
| *Adipor1* | AATGGGGCTCCTTCTGGTAAC | GCAGACCTTATACACGAACTCC |
| *Adipor2* | CCTTTCGGGCCTGTTTTAAGA | GAGTGGCAGTACACCGTGTG |

*Continued on next page*

| | | | |
|---|---|---|---|
| *Pgc1a* | AACCAGTACAACAATGAGCCTG | AATGAGGGCAATCCGTCTTCA | |
| **CRISPRsgRNA** | | | |
| Target | | | |
| *Bmal1* KO | CCCACAGTCAGATTGAAAAG | | |
| **Cloning** | | | |
| Target | Forward | Reverse | References |
| *Npy* promoter | GTCCAGGAGGTGATGAA CCTATGTTCTTTATGG | GCGCCCCTGTCCCA GTTGATCCTGGC | *Fick et al., 2010* |
| **ChIP** | | | |
| *Bmal1* RORE | TTGGGCACAGCGATTGGTGG | TCCGGCGCGGGTAAACAGG | |
| *Bmal1* 3'UTR | CTGTGTGCATCGGACAGTC | CGAAGCCACCATCTGAAAC | |

# Acknowledgements

The authors would like to thank Prof. Steve Brown for donating the *Bmal1-luc* plasmid and Nadine Oster and Ludmila Skrum for expert technical assistance.

# Additional information

## Funding

| Funder | Grant reference number | Author |
|---|---|---|
| Deutsche Forschungsgemeinschaft | GRK-1957 | Henrik Oster |
| Deutsche Forschungsgemeinschaft | OS353-7/1 | Henrik Oster |
| Volkswagen Foundation | Lichtenberg Professorship | Henrik Oster |
| Deutsche Forschungsgemeinschaft | OS353-10/1 | Henrik Oster |

The funders had no role in study design, data collection and interpretation, or the decision to submit the work for publication.

## Author contributions

Anthony H Tsang, Christiane E Koch, Jana-Thabea Kiehn, Cosima X Schmidt, Formal analysis, Investigation, Writing - review and editing; Henrik Oster, Conceptualization, Formal analysis, Supervision, Validation, Writing - original draft, Writing - review and editing

## Author ORCIDs

Henrik Oster (iD) https://orcid.org/0000-0002-1414-7068

## Ethics

Animal experimentation: All animal experiments were done after ethical assessment by the institutional animal welfare committee and licensed by the Office of Consumer Protection and Food Safety of the State of Lower Saxony (33.12.42502-04-12/0893, 33.14-42502-04-11/0604 and 33.9-42502-04-12/0748) or the Ministry of Agriculture of the State of Schleswig-Holstein (V 242-7224.122-4 (132-10/13)) in accordance with the German Law of Animal Welfare (TierSchG).

## Decision letter and Author response

Decision letter https://doi.org/10.7554/eLife.55388.sa1
Author response https://doi.org/10.7554/eLife.55388.sa2

## Additional files

### Supplementary files

- Transparent reporting form

### Data availability

All data generated or analysed during this study are included in the manuscript and supporting files. Source data files have been provided for Figures 1 to 8.

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
