## [Decision Letter]

**Acceptance summary:**

This manuscript provides a mechanism by which energy status in the periphery is transmitted to the brain to regulate the timing of food intake. Given the prevalence of metabolic disorders in modern society, it is important to understand how these might arise from disruption of the processes investigated here.

**Decision letter after peer review:**

Thank you for sending your article entitled "An adipokine feedback regulating diurnal food intake rhythms in mice" for peer review at *eLife*. Your article has been evaluated by three peer reviewers, and the evaluation has been overseen by a Reviewing Editor and Catherine Dulac as the Senior Editor.

The reviewers found your manuscript of interest, but felt that it lacked sufficient mechanistic detail. Given that some findings reported here are known , the novel aspects, which pertain to the rhythmic anorexigenic action of ADIPOQ presumably through the MBH clock, should be further developed. For instance, it is unclear how adiponectin affects the MBH clock – is this through BMal1 and, if yes, how is BMal1 induced-- and whether BMal1/the MBH clock is required for anorexigenic effects of ADIPOQ. Ideally, both these questions would be addressed, but at the very least, we would like to see experiments determining whether *Bmal*1 is required for anorexigenic effects.

Also, it is very important that you cite and discuss the findings of Paschos et al., 2012, which showed that knockout of the adipose tissue clock affects feeding rhythms in a similar manner as reported here, but via lipids rather than adiponectin.

Essential revisions:

1) One major shortcoming in this study is that it includes only male mice. While it would be prohibitively time consuming to repeat all of their experiments in female mice, it would be nice to see at least some of the findings evaluated in female mice, perhaps the impact of *Adipoq* deletion on daily patterns of activity and food intake or the impact of timed ADIPOQ infusion in mice on a HFD. Regardless, this limitation must be stated clearly. Also, gender should be indicated in the figure legends also, and not just the Materials and methods.

2) In the Abstract, there are six claims which are mostly very well supported. The only one that is less solid is the statement that ADIPOQ regulates MBH clocks via *AdipoR1*-mediated upregulation of *Bmal1*. They clearly show that ADIPOQ regulates MBH via *AdipoR1* and that the expression of *Bmal1* is altered by ADIPOQ and by *AdipoR1*. It is not clear that the effect of ADIPOQ on MBH clocks is caused by altered *Bmal1* expression.

3) Figure 1 examines the impact of restricting food access to a four-hour window during either the late day or late night hours. How does overall food intake compare to ad libitum in these groups?

4) The text describing Figure 1I-L states that the impact of HFD conditions are similar for plasma ADIPOQ and for MBH expression of *Adipoq*, *Adipor2*, *Npy* and *Agrp*. However, the data suggest that the impact of HFD on *Adipor2* expression is different from that on *Adipoq*, *Npy*, and *Agrp* which do look similar. I confess I am a bit confused by the way the statistics are presented in Figure 1. The authors appropriately used two-way ANOVA to analyze these data but I don't understand the way they present the statistical results.

5) Figure 2 describes activity, feeding patterns and gene expression in mice harboring germline deletion of *Adipoq*. The text states that the activity patterns were "largely comparable" but the data in Figure 1A shows that the mutant mice are more active early in the night in LD conditions. If anything, this makes the altered food intake more striking so it does not diminish those results but it should be accurately described.

6) The lack of effect of constant ADIPOQ infusion is intriguing given the strong impact of timed infusion. This should be discussed.

7) The Materials and methods state that the substance solutions in the pumps were refreshed every 7-8 days but I could not find a description of the duration of infusion.

8) I could not find a description of what time of day the i.v. injection of ADIPOQ was performed after which MBH expression of *Bmal1* was measured (Figure 5). Given the apparent phase-shifting effect of ADIPOQ, this information is critical to distinguish between direct activation of *Bmal1* expression and altered expression caused by a phase shift. Similarly, it would be helpful to see the impact of ADIPOQ on *Bmal1* expression in unsynchronized N44 cells. For these reasons as well as the remaining effect of Adn injection on *Npy* and *Agrp* expression in *Bmal1*-deficient cells and MBH (Figure 5E, F), the conclusions regarding a BMAL1-mediated mechanism should be softened.

9) Similarly, to make a claim that ADIPOQ-driven clock resetting does not require changes in *Per* expression, *per1*, *per2*, and *per3* transcripts should be measured.

10) It is a bit odd that the peak of *Npy* and *Agrp* expression in ad libitum fed animals peaks (ZT12) before the peaks of ADIPOQ in plasma and antiphase to *Adipor1* expression in MBH. Do the authors have any ideas to explain this?

11) In Figure 3, *Bmal1*, *Npy*, and *Agrp* expression should also be measured in the WT and MT/AR constant groups.

---

## [Author Response]

The reviewers found your manuscript of interest, but felt that it lacked sufficient mechanistic detail. Given that some findings reported here are known , the novel aspects, which pertain to the rhythmic anorexigenic action of ADIPOQ presumably through the MBH clock, should be further developed. For instance, it is unclear how adiponectin affects the MBH clock-- is this through BMal1 and, if yes, how is BMal1 induced-- and whether BMal1/the MBH clock is required for anorexigenic effects of ADIPOQ. Ideally, both these questions would be addressed, but at the very least, we would like to see experiments determining whether Bmal1 is required for anorexigenic effects.

We now include additional data on the regulation of *Bmal1* by ADIPOQ in N44 cells and the MBH showing that PGC1a mediates induction of *Bmal1* expression and clock resetting by ADIPOQ/*AdipoR1* (new Figures 4-6). Complementary to the wild-type data in the original Figure 6 (now Figure 8) we have added new data on the effect of ADIPOQ infusion in *Bmal1* knockout mice – which has no effect on body weight regulation independent of the treatment schedule (new Figure 8). We also added data on the phenotype of female *Adipoq* deficient mice (new Figure 3 with two supplementary figures).

Also, it is very important that you cite and discuss the findings of Paschos et al., 2012, which showed that knockout of the adipose tissue clock affects feeding rhythms in a similar manner as reported here, but via lipids rather than adiponectin.

We added a paragraph to the Discussion on this topic. In brief, we believe that both pathways may be necessary for normal rhythmic food intake. It would be interesting to study their interaction under different metabolic conditions, but we feel that this is beyond the scope of this manuscript.

Essential revisions:1) One major shortcoming in this study is that it includes only male mice. While it would be prohibitively time consuming to repeat all of their experiments in female mice, it would be nice to see at least some of the findings evaluated in female mice, perhaps the impact of Adipoq deletion on daily patterns of activity and food intake or the impact of timed ADIPOQ infusion in mice on a HFD. Regardless, this limitation must be stated clearly. Also, gender should be indicated in the figure legends also, and not just the Materials and methods.

We have a data set on circadian rhythms in female *Adipoq* knockout mice (new Figure 3 and Figure 3—figure supplements 1 and 2). Unfortunately, we have no data on the anorexigenic effects of AdipoRon infusion in females. We have stated this limitation in the revision. We also added gender information to the figure legends.

2) In the Abstract, there are six claims which are mostly very well supported. The only one that is less solid is the statement that ADIPOQ regulates MBH clocks via AdipoR1-mediated upregulation of Bmal1. They clearly show that ADIPOQ regulates MBH via AdipoR1 and that the expression of Bmal1 is altered by ADIPOQ and by AdipoR1. It is not clear that the effect of ADIPOQ on MBH clocks is caused by altered Bmal1 expression.

We added more mechanistic data on the regulation of *Bmal1* by ADIPOQ in N44 cells and the MBH suggesting that PGC1a mediates induction of *Bmal1* expression and clock resetting by ADIPOQ/*AdipoR1* (new Figures 5-7). Of note, PGC1a-mediated *Bmal1* induction has been shown for liver before (Liu et al., 2007).

3) Figure 1 examines the impact of restricting food access to a four-hour window during either the late day or late night hours. How does overall food intake compare to ad libitum in these groups?

Animals ate slightly, but not significantly less calories under RF conditions. Differences were higher during the first days of the regimen, but animals were sacrificed after two weeks when body weight had largely recovered. We have added this information to the manuscript.

4) The text describing Figure 1I-L states that the impact of HFD conditions are similar for plasma ADIPOQ and for MBH expression of Adipoq, Adipor2, Npy and Agrp. However, the data suggest that the impact of HFD on Adipor2 expression is different from that on Adipoq, Npy, and Agrp which do look similar. I confess I am a bit confused by the way the statistics are presented in Figure 1. The authors appropriately used two-way ANOVA to analyze these data but I don't understand the way they present the statistical results.

We thank the reviewer for this observation. We have revised the wording as *AdipoR1/2* regulation is different from the other analyzed factors. We do not think that this compromises our conclusions as we do not claim that a regulation of AipoR expression itself is part of the ADIPOQ-MBH clock-feeding signaling pathway. We modified the text accordingly.

5) Figure 2 describes activity, feeding patterns and gene expression in mice harboring germline deletion of Adipoq. The text states that the activity patterns were "largely comparable" but the data in Figure 1A shows that the mutant mice are more active early in the night in LD conditions. If anything, this makes the altered food intake more striking so it does not diminish those results but it should be accurately described.

We have revised the manuscript accordingly.

6) The lack of effect of constant ADIPOQ infusion is intriguing given the strong impact of timed infusion. This should be discussed.

We have added a paragraph on this in the Discussion. In brief, we think that ADIPOQ has a phasic, not so much a tonic impact on MBH clock function and food intake rhythms.

7) The Materials and methods state that the substance solutions in the pumps were refreshed every 7-8 days but I could not find a description of the duration of infusion.

The infusion schedule is depicted in new Figures 4A and 8A. We also altered the indicator of the infusion in the body weight curves (new Figure 8E, H) and refer to the A panels in the Materials and methods section to make this clear.

8) I could not find a description of what time of day the i.v. injection of ADIPOQ was performed after which MBH expression of Bmal1 was measured (Figure 5). Given the apparent phase-shifting effect of ADIPOQ, this information is critical to distinguish between direct activation of Bmal1 expression and altered expression caused by a phase shift.

We apologize for this omission. Injections were done at ZT6 and animals sacrificed 3 h later. We have added this information to the manuscript.

Similarly, it would be helpful to see the impact of ADIPOQ on Bmal1 expression in unsynchronized N44 cells.

We show this in Figure 6A. We also now provide qPCR data in Figure 6B.

For these reasons as well as the remaining effect of Adn injection on Npy and Agrp expression in Bmal1-deficient cells and MBH (Figure 5E, F), the conclusions regarding a BMAL1-mediated mechanism should be softened.

We agree that we cannot completely rule out that additional resetting mechanisms through other clock components than *Bmal1* may also be involved. We have stated this in the Discussion.

9) Similarly, to make a claim that ADIPOQ-driven clock resetting does not require changes in Per expression, per1, per2, and per3 transcripts should be measured.

We now provide these data in Figure 6C-F.

10) It is a bit odd that the peak of Npy and Agrp expression in ad libitum fed animals peaks (ZT12) before the peaks of ADIPOQ in plasma and antiphase to Adipor1 expression in MBH. Do the authors have any ideas to explain this?

While NPY/AGRP and ADIPOQ plasma levels roughly cycle in phase, the anti-phasic expression of *AdipoR1* appears odd. We have no good explanation for this at the moment, other than *AdipoR1* levels may not be rate-limiting for the MBH clock-resetting effects of ADIPOQ. We have mentioned this in the Discussion.

11) In Figure 3, Bmal1, Npy, and Agrp expression should also be measured in the WT and MT/AR constant groups.

We have added these data (new Figure 4D-F).